# A chronic *Acinetobacter baumannii* pneumonia model to study long-term virulence factors, antibiotic treatments, and polymicrobial infections

Clay D. Jackson-Litteken[1,5], Gisela Di Venanzio[1], Manon Janet-Maitre [1], Ítalo A. Castro [1,2], Joseph J. Mackel[3], Leslie D. Wilson[4], David A. Rosen [1,3], Carolina B. López [1,2] & Mario F. Feldman [1] ✉

*Acinetobacter baumannii* causes prolonged infections that disproportionately affect immunocompromised populations. Our understanding of *A. baumannii* respiratory pathogenesis relies on an acute murine infection model with limited clinical relevance that employs an unnaturally high number of bacteria and requires assessment of bacterial load at 24-36 h post-infection. Here, we demonstrate that low intranasal inoculums in *tlr4* mutant mice allows for infections lasting at least 3 weeks. Using this "chronic infection model" we determine the adhesin InvL is a virulence factor required during later stages of infection, despite being dispensable in the early phase. We also demonstrate that the chronic model enables distinction between antibiotics that, although initially reduce bacterial burden, either lead to clearance or result in the formation of potential bacterial persisters. To illustrate how our model can be applied to study polymicrobial infections, we inoculate mice with an active *A. baumannii* infection with *Staphylococcus aureus* or *Klebsiella pneumoniae*. We find that *S. aureus* exacerbates infection, while *K. pneumoniae* enhances *A. baumannii* clearance. In all, the chronic model overcomes some limitations of the acute pulmonary model, expanding our capabilities to study *A. baumannii* pathogenesis and lays the groundwork for the development of similar models for other opportunistic pathogens.

*Acinetobacter baumannii* is a Gram-negative opportunistic pathogen that causes diverse infections including pneumonia, urinary tract infection (UTI), bone and soft tissue infection, and septicemia[1–5]. While becoming an increasingly more common cause of community-acquired infections, *A. baumannii* still primarily causes hospital-

acquired infections in critically ill and immunocompromised patients, ~25% of which are polymicrobial[6–11]. These infections are associated with an alarming mortality rate, up to 80% in some populations, largely owing to extremely high rates of multi-drug resistance[8,12,13]. Notably, *A. baumannii* isolates exhibit the highest

[1]Department of Molecular Microbiology, Washington University School of Medicine, Saint Louis, MO, USA. [2]Center for Women's Infectious Diseases Research, Washington University School of Medicine, Saint Louis, MO, USA. [3]Department of Pediatrics, Division of Infectious Diseases, Washington University School of Medicine, Saint Louis, MO, USA. [4]Division of Comparative Medicine, Research Animal Diagnostic Laboratory, Washington University School of Medicine, Saint Louis, MO, USA. [5]Present address: Department of Microbiology and Immunology, University of Arkansas for Medical Sciences, Little Rock, AR, USA. ✉e-mail: mariofeldman@wustl.edu

rates of multi-drug resistance of all Gram-negative pathogens, leading the World Health Organization to classify the bacterium at its highest priority for research and development of new treatments[13,14]. There is consequently an urgent need to better understand the virulence mechanisms employed by *A. baumannii* to guide the development of novel therapeutic approaches to combat infections.

While *A. baumannii* can cause a variety of infections, it is most commonly associated with pneumonia[4,15]. In fact, *A. baumannii* causes up to 10% of all hospital-acquired pneumonia (HAP) cases in the United States, highlighting its importance in clinical settings[16,17]. Despite this, little is known regarding the pathogenesis of this bacterium in the respiratory tract[18]. A major hindrance in the ability to investigate *A. baumannii* pneumonia is the lack of available clinically-relevant murine infection models. This is, in large part, due to the low virulence of most strains in immunocompetent mice. This is a shared feature among many pathogens that commonly cause HAP, including *Pseudomonas aeruginosa* and *Staphylococcus aureus*, for which animal models closely mimicking human infection are not available[19,20]. An acute infection model requiring a very high, and rather artificial, inoculum of $10^8$–$10^9$ bacteria introduced intranasally or intratracheally is most often used to investigate these pathogens[10,20,21]. Wild-type (WT) mice will typically either succumb to infection or clear the organism by 72 h, thus requiring early readouts of infection such as bacterial pulmonary titers at 24–36 h. While this model may serve as a useful tool to study pathogenesis early during infection, the quick bacterial clearance does not allow for the study of bacterial virulence mechanisms at later timepoints. Importantly, *A. baumannii* respiratory infection in humans results in an average length of hospital stay of ~30 days, and this number is much higher in cases caused by multi-drug resistant strains, highlighting the need for a long-term infection model[22,23]. In this pursuit, some laboratories have used antibody or cyclophosphamide treatments to render mice neutropenic[24–29]. These treatments initially make mice more susceptible to *A. baumannii* infection, enabling the study of bacterial pathogenesis up to 7 d post-infection (dpi) using lower inoculums (~$10^7$ bacteria). However, these models do not achieve stable neutropenia in mice which leads to clearance of infection. To maintain neutropenia over longer periods, multiple injections would be necessary, which can lead to fluctuating neutrophil levels, thereby altering the overall course of disease[30–34]. In all, there is an urgent need for alternative infection models to study bacterial pathogenesis during long-term infection by relevant clinical isolates.

Previous reports have used genetically immunocompromised mice to study the role of the host immune response to *A. baumannii* infection. One example is mice carrying a mutation in toll-like receptor 4 (*tlr4*). TLR4 recognizes the lipid A moiety of bacterial lipopolysaccharide (LPS) and lipooligosaccharide (LOS), the main component of the outer membrane of most Gram-negative bacteria[30–32]. The recognition of lipid A by TLR4 triggers a signaling cascade through MyD88- or TRIF-dependent pathways, resulting in increased inflammatory cytokine and type 1 interferon production, respectively[33]. The role of TLR4 during *A. baumannii* infection has been examined in murine septicemia, acute pneumonia, UTI, and catheter-associated UTI (CAUTI) models[34–36]. In the acute pneumonia model, Knapp et al. showed that *tlr4* mutant mice had increased *A. baumannii* CFU in the lungs with reduced inflammatory cytokines compared to WT mice[35,37]. Using a bloodstream infection model, Lin et al. demonstrated that WT C3H/FeJ and *tlr4* mutant C3H/HeJ mice had similar bacterial burdens[34,38]. However, all WT mice succumbed to infection by day 4, whereas all *tlr4* mutant mice survived. This could be attributed to WT mice experiencing septic shock associated with increased inflammatory cytokines. Finally, in a UTI model, our laboratory found that *tlr4* mutant C3H/HeJ mice were more susceptible to infection than WT C3H/HeN mice[36]. Moreover, we found that C3H/HeJ mice in the UTI model formed small intracellular populations in urothelial cells referred to as *Acinetobacter baumannii* intracellular reservoirs (ABIRs),

which could seed a recurrent infection upon catheterization at higher rates relative to WT mice. In addition to playing a significant role during murine infection, TLR4 is relevant in clinical settings as well. In fact, numerous studies have identified links between *tlr4* polymorphisms and infection outcomes from *A. baumannii* pneumonia in humans[39–41]. In all, these studies demonstrate the key role of TLR4 in controlling *A. baumannii* infection and disease progression and highlight the clinical relevance of the associated signaling cascade.

In this work we describe a novel murine model of *A. baumannii* pneumonia that employs *tlr4* mutant mice and low bacterial inoculums ($10^5$ bacteria). Using this model, we show that *A. baumannii* can establish chronic infection lasting several weeks, which is more clinically relevant than shorter-term models. We additionally demonstrate that our model enables the discovery of virulence factors not detectable in the acute infection model. Finally, we illustrate how our model can be employed to assess the efficacy of antibiotics over the course of infection and investigate polymicrobial infections.

## Results

### tlr4 mutant mice are susceptible to chronic infection at low inoculums

To assess if *tlr4* mutant mice could serve as permissive hosts for long-term respiratory infection, we performed intranasal inoculations of WT (C3H/HeN) and *tlr4* mutant (C3H/HeJ) mice with a range of inoculums of a modern *A. baumannii* respiratory isolate, G636, and sacrificed groups of mice every 3 days starting at 24 h post-infection (hpi). At the highest inoculum of $10^8$ colony forming units (CFU), WT mice cleared infection by day 4, consistent with previously published results using the acute pulmonary infection model (Fig. 1A)[21]. *tlr4* mutant mice infected with $10^8$ CFU also cleared infection relatively early after inoculation, with most mice having no detectable bacteria in the lungs by ~7 dpi. Strikingly, while WT mice infected with a lower dose of $10^5$ bacteria cleared infection after 1 day, *tlr4* mutant mice maintained detectable bacteria in the lungs out to the latest timepoint tested in this experiment, 19 dpi (Fig. 1D). A subsequent extended experiment revealed that some *tlr4* mutant mice infected with $10^5$ G636 maintain detectable levels of CFU in the lungs even out to 3–4 weeks pi (Fig. S1). Despite this long infection course in *tlr4* mutant mice at this dose, dissemination to distal organs was rarely detected. This is consistent with the clinical manifestations of non-ventilator *A. baumannii* pneumonia, as less than 20% of patients will develop subsequent bacteremia[42]. Notably, by employing confocal microscopy, we were able to visualize bacteria in *tlr4* mutant mice infected with the $10^5$ inoculum; at early timepoints (4 hpi and 2 dpi), bacteria were identified inside cells in the bronchoalveolar lavage fluid (BALF), as well as extracellularly, consistent with our previously published results in the acute infection model (Fig. S2)[43]. We also screened other inoculums of G636 in *tlr4* mutant mice above and below $10^5$ CFU (Fig. S3). As inoculums increased above $10^5$ CFU however, we noted quicker time to bacterial clearance; mice infected with $10^6$ and $10^7$ CFU began clearing infection by 10 and 13 dpi, respectively. *tlr4* mutant mice infected with inoculums lower than $10^5$ CFU also gave less optimal results; the $10^4$ inoculum resulted in slightly earlier clearance, and the $10^3$ inoculum resulted in inconsistent infection. Consequently, we selected the $10^5$ CFU inoculum as the optimal dose for studying long-term infection.

We next evaluated whether a second modern *A. baumannii* respiratory isolate, G654, exhibits similar behavior to G636 (Figs. 1E and S1). As with G636, detectable levels of bacteria were present in the lungs out to 19 dpi with the $10^5$ inoculum in *tlr4* mutant mice, while WT mice cleared this inoculum within 1 day. At the higher inoculum of $10^8$ CFU, G654 infection was inconsistent in *tlr4* mutant mice and infection was cleared early after infection in WT mice (Fig. 1B). Finally, we screened a third modern respiratory isolate, G803, which is phylogenetically and geographically unrelated to G636 and G654, for its ability to establish long-term infection with a $10^5$ inoculum

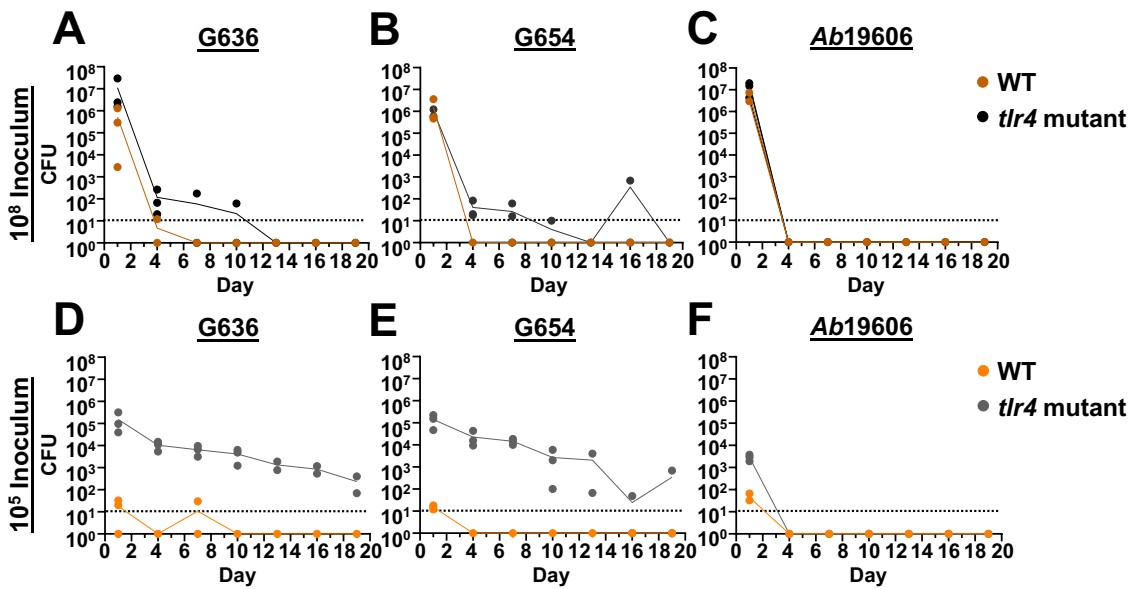

**Fig. 1 | Low inoculums of modern respiratory _A. baumannii_ clinical isolates result in chronic lung infection in _tlr4_ mutant mice.** Groups of female C3H/HeN (WT) or C3H/HeJ (_tlr4_ mutant) mice were intranasally inoculated with $10^8$ G636 (**A**), $10^8$ G654 (**B**), $10^8$ Ab19606 (**C**), $10^5$ G636 (**D**), $10^5$ G654 (**E**), or $10^5$ Ab19606 (**F**). Beginning at 24 hpi, groups of mice were sacrificed every 3 days, and bacteria in the lungs were quantified. Each data point indicates an individual mouse, and the connecting line intersects each timepoint at the mean. The limit of detection (10 CFU) is indicated by the dashed line. Source data are provided as a Source Data file.

in _tlr4_ mutant mice. G803 was also able to establish long-term infection, as CFU were recovered from the lungs up to 16 dpi (Table S1 and Fig. S4A).

Given possible genotypic and phenotypic differences between older and modern isolates, we also screened commonly employed older, lab-domesticated strains (Ab19606 and Ab17978) to see if these can similarly persist in _tlr4_ mutant mice at low inoculums. In contrast to infection kinetics observed with the modern isolates, Ab19606 infection was cleared after 1 day regardless of inoculum size or mouse background (Fig. 1C, F). Alternatively, Ab17978 was able to establish persistent infection in _tlr4_ mutant mice with the low inoculum out to the latest timepoint tested for this strain, 7 dpi, with similar kinetics to those of the modern respiratory isolates tested (Fig. S4B). This finding that Ab17978 and Ab19606 have differential infection phenotypes highlight strain-dependent differences among _A. baumannii_ isolates and is in agreement with other infection models classifying Ab19606 as a low virulence strain[44,45].

In all, these results indicate that 3 modern _A. baumannii_ respiratory isolates can cause infection out to ~3 weeks pi at lower and likely more clinically-relevant inoculums than previously used in the literature. Importantly, this infection duration noted here with low inoculums of modern respiratory isolates in _tlr4_ mutant mice is the longest reported for _A. baumannii_ in any animal model to date. We therefore chose to further characterize these conditions as a model to study pulmonary pathogenesis, referred to hereafter as the "chronic respiratory infection model."

**Lower _A. baumannii_ inoculums result in a decreased immune response in _tlr4_ mutant mice**

Given the unexpected result that _tlr4_ mutant mice exhibit chronic infection at lower inoculums, while WT and _tlr4_ mutant mice clear infection at higher inoculums, we sought to characterize the host immune response in these different conditions. We intranasally infected groups of WT and _tlr4_ mutant mice with $10^5$ or $10^8$ bacteria or mock infected them with phosphate-buffered saline (PBS). Then, at early timepoints of 4 hpi and 2 dpi and a later timepoint of 7 dpi, BALF was collected for immune cell quantification (Figs. 2 and S5). Regarding alveolar macrophages (AMs), initial reductions were noted at early

timepoints with the large inoculum. However, AM numbers rebounded by 2 dpi and minimal differences were noted after this timepoint (Fig. 2A–C). In WT mice, the number of polymorphonuclear leukocytes (PMNs) was increased with the higher inoculum relative to lower and mock inoculums at every timepoint (Fig. 2D–F). This result at 4 and 48 hpi aligns with previous reports comparing _A. baumannii_ acute murine pulmonary infection to mock infection[46–53]. However, the result at 7 dpi that neutrophils were still significantly elevated in WT mice relative to mock-infected controls diverges from some previous reports that indicate that neutrophil levels of WT mice regress to those of mock-infected mice following 48 hpi[47,50]. However, other reports showed still increasing levels of neutrophils at 72 hpi, with one showing a decrease only after this timepoint[52,53]. These differences can likely be attributed to use of different mouse (C3H/HeN, C57BL/6, A/J) or _A. baumannii_ (G636, ATCC17961, ATCC BAA-1605, A112-II-a) strains in these previous papers. However, neutrophil levels are still decreasing following 48 h of high inoculum infection in C3H/HeN mice in line with these previous studies (~2 $\log_{10}$ between 2 and 7 dpi). At the higher inoculum, WT mice also had increased PMNs relative to _tlr4_ mutant mice early during infection in line with previous results[35]. Interestingly, at the lower inoculum, while PMN counts trended higher at 4 hpi for WT mice relative to _tlr4_ mutant mice, no significant differences were noted between these groups at any timepoint. This result suggests that, despite neutrophil influx being a predominant mechanism of _A. baumannii_ clearance, PMN numbers alone may not account for differences in clearance between WT and _tlr4_ mutant mice at the lower inoculum[47,50,53–57].

Given the established importance of PMNs in _A. baumannii_ infection, we decided to further assess the role of PMNs in the chronic infection model. We rendered _tlr4_ mutant mice neutropenic with cyclophosphamide prior to infection with $10^5$ G636 via a previous established protocol by Manepalli et al.[29] (Fig. S6). Neutropenic _tlr4_ mutant mice had an initial bacterial burden of $10^7$ CFU at 1 dpi. This is 100-fold higher than numbers seen in the chronic infection model at the same timepoint (see Fig. 1). As expected though, as neutrophil numbers rebound following cyclophosphamide treatment (3–4 days post-treatment), bacterial CFU began to approach levels similar to those observed in the chronic infection model[29,58,59]. This result implies that while PMNs are not effective in efficiently clearing infection in the

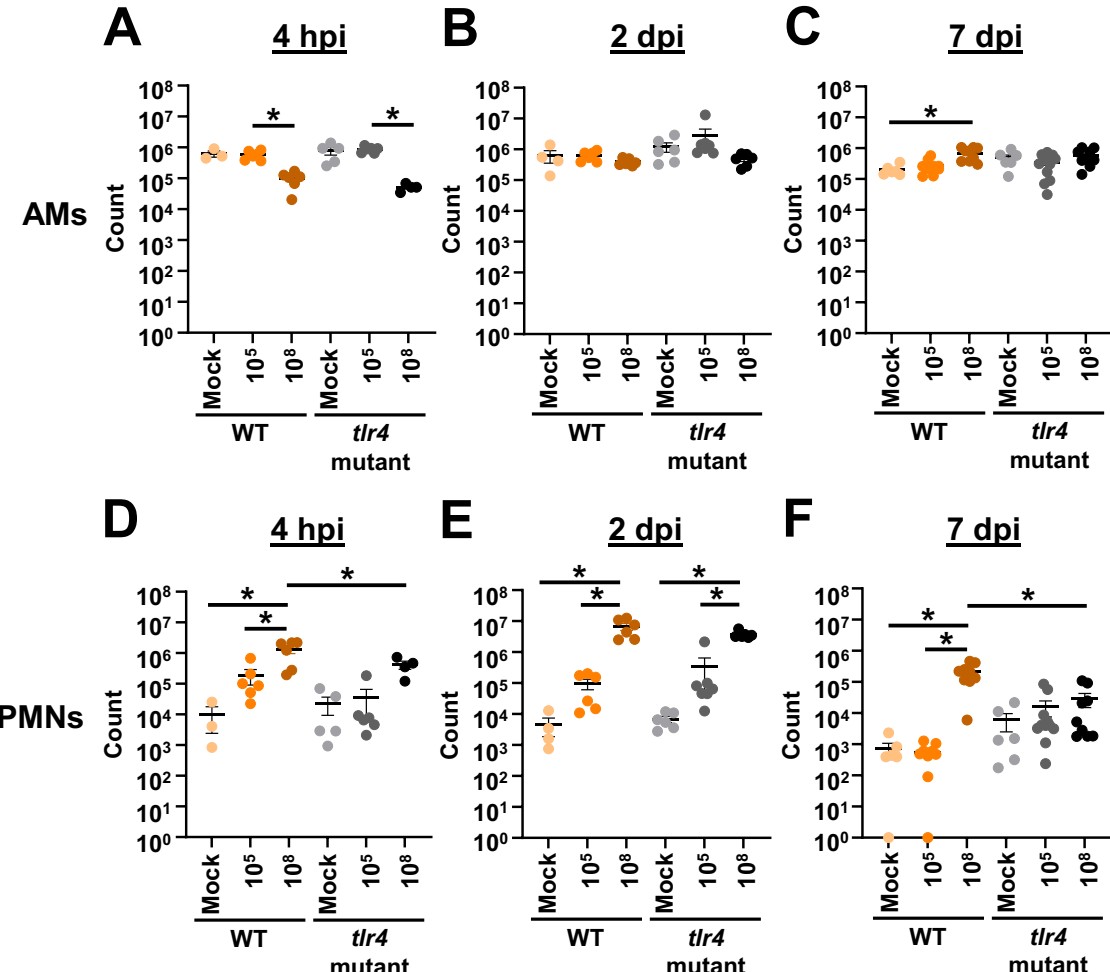

**Fig. 2 | Lower intranasal *A. baumannii* inoculums result in reduced lung neutrophil influx.** Groups of female C3H/HeN (WT) or C3H/HeJ (*tlr4* mutant) mice were intranasally inoculated with $10^5$ G636, $10^8$ G636, or mock inoculated with PBS. At 4 h (**A**, **D**), 2 d (**B**, **E**), and 7 d (**C**, **F**) pi, alveolar macrophages (AMs) (**A**–**C**) and polymorphonuclear leukocytes (PMNs) (**D**–**F**) in the BALF were enumerated by flow cytometry. Shown are pooled results from at least two independent experiments, and each data point represents an individual mouse. The horizontal line represents the mean, and the standard error of the mean (SEM) is indicated by error bars. *$p < 0.05$; two-way analysis of variance (ANOVA), Tukey's test for multiple comparisons. Shown are statistically different significances between different strains at the same inoculum and between different inoculums given to the same strain. Source data and statistical test details are provided in the Source Data file. *p* values: 4 hpi AMs WT, $10^5$ vs. Mock = 0.0242; 4 hpi AMs *tlr4* mutant $10^5$ vs. $10^8$ = 0.0003; 7 dpi AMs, WT Mock vs. $10^8$ = 0.0272; 4 hpi PMNs, WT Mock vs. $10^8$ = 0.0037; 4 hpi PMNs, WT $10^5$ vs. $10^8$ = 0.0020; 4 hpi PMNs, WT $10^8$ vs. tlr4 mutant $10^8$ = 0.0402; 2 dpi PMNs, WT Mock vs. $10^8$ = < 0.0001; 2 dpi PMNs, WT $10^5$ vs. $10^8$ = < 0.0001; 2 dpi PMNs, *tlr4* mutant Mock vs. $10^8$ = 0.0190; 2 dpi PMNs, *tlr4* mutant $10^5$ vs. $10^8$ = 0.0313; 7 dpi PMNs, WT Mock vs. $10^8$ = < 0.0001; 7 dpi PMNs, WT $10^5$ vs. $10^8$ = < 0.0001; 7 dpi PMNs, WT $10^8$ vs. *tlr4* mutant $10^8$ = < 0.0001.

chronic infection model, they still maintain a role in controlling bacterial proliferation.

To further evaluate the host response, we quantified 13 common inflammatory cytokines in the BALF (Table S2). At the higher inoculum, WT and *tlr4* mutant mice exhibited significantly increased levels of IL-1α, IFN-γ, TNF-α, MCP-1, IL-1β, IL-6, and IL-17A early during infection relative to the lower inoculum while levels dissipated by 7 dpi, consistent with bacterial clearance (see Fig. 1A). WT mice infected with the high inoculum had significantly increased levels of IFN-β relative to *tlr4* mutant mice at 4 hpi and increased levels of IL-1α, IFN-γ, TNF-α, MCP-1, IL12-p70, IL-1β, IL-6, IL-27, and IL17A at 2 dpi, likely leading to the earlier clearance observed. At the lower inoculum, although WT mice clear infection within nearly 24 h and *tlr4* mutant mice maintain infection out to at least 3 weeks, minimal significant differences in inflammatory cytokines were observed (see Fig. 1D). In fact, the only significant difference noted was the increased levels of GM-CSF at 4 hpi in WT mice relative to *tlr4* mutant mice. Other inflammatory cytokines that trended higher at early timepoints in WT mice at the lower inoculum include the inflammasome-associated cytokines IL-1α and IL-1β, as well as TNF-α and IL-6. These elevated levels of inflammatory cytokines early during infection in WT mice could possibly account for the earlier clearance. Later during infection, however, *tlr4* mutant mice had elevated, albeit not significantly higher, amounts of TNF-α, IL-1α, and IL-6 relative to WT mice, consistent with persistent infection.

### The chronic respiratory infection model results in alveolitis and airway epithelium attenuation

We next assessed if the chronic respiratory infection model is associated with lung pathology. *tlr4* mutant mice were infected with $10^5$ G636 or mock infected with PBS, and mice were sacrificed at 4 hpi, 2 dpi, 7 dpi, 14 dpi, and 21 dpi. Lungs were then sectioned, stained with hematoxylin and eosin (H&E), and scored for pathological changes by a licensed veterinary pathologist (Fig. 3). The chronic model resulted in a significant increase in alveolitis at 2 dpi (Fig. 3A) and airway epithelium attenuation at 7 and 21 dpi (Fig. 3B) relative to mock-infected mice. Moreover, trends toward higher levels of these phenotypes were observed at most timepoints (Fig. 3C–E).

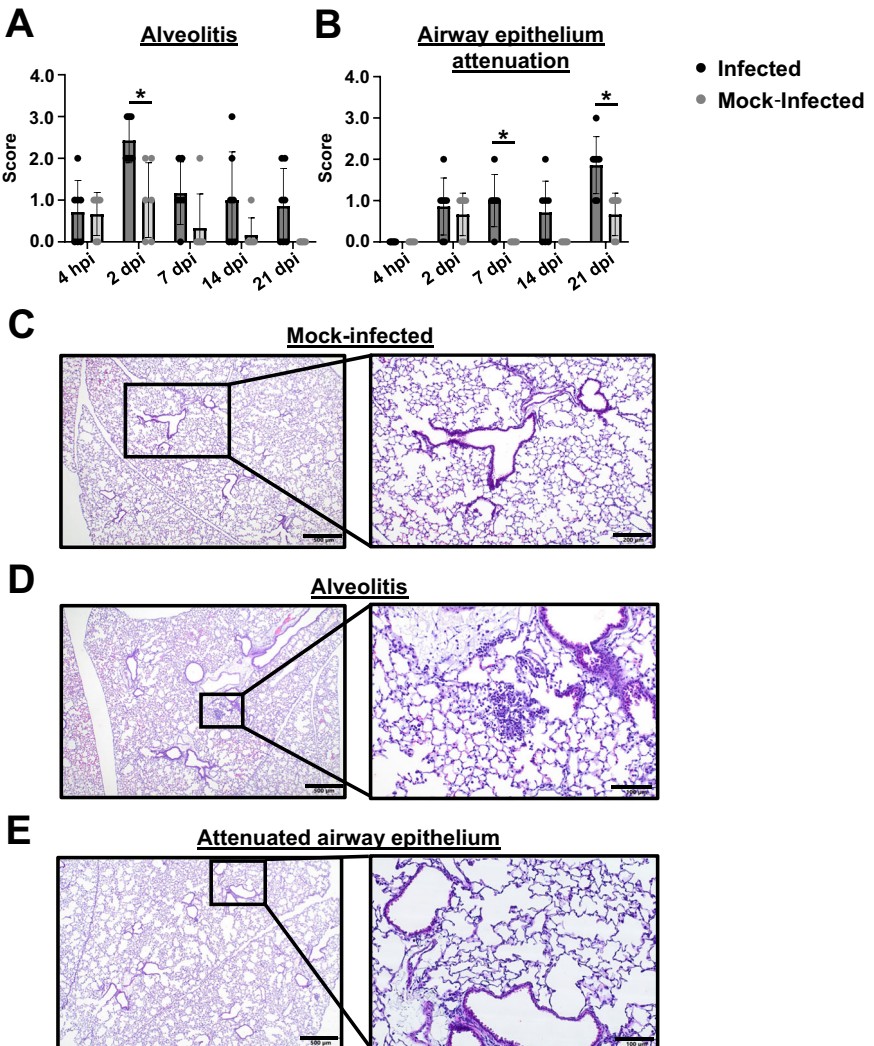

**Fig. 3 | The chronic respiratory infection model results in lung pathology.** Groups of female C3H/HeJ (*tlr4* mutant) mice were inoculated with 10⁵ G636 or mock-inoculated with PBS, and at 4 hpi, 2 dpi, 7 dpi, 14 dpi, and 21 dpi, lungs slices were prepared, H&E stained, and scored for alveolitis (**A**) or airway epithelium attenuation (**B**). Shown are three biological replicates with 2/3 mice/replicate ($n = 6$–7). The bar represents the mean, each mouse is indicated by a dot, and the SEM is indicated by error bars. *$p < 0.05$; two-way ANOVA, Bonferroni's test for multiple comparisons. Representative tissue sections of mock-infected tissue (**C**) and alveolitis (**D**) and attenuated airway epithelium (**E**) in infected tissue are shown. Scale bars = 500 µm and inset scale bars = 200 µm.

We also examined other potential inflammatory pathologies in these experiments, revealing alveolar histiocytosis, perivascular infiltrate, peribronchiolitis, bronchoalveolar lymphoid tissue (BALT), and pleuritis in both infected and mock-infected animals (Fig. S7). These findings were not surprising, however, as *tlr4* mutation results in changes in lung development and morphology, such as airspace enlargement and emphysema[60]. Moreover, naïve C3H/HeJ mice have been shown to have basal increased levels of inflammation relative to wild-type C3H background mice, including increased terminal airspace, inflammatory cytokines, and immune cells possibly due to colonization by environmental bacteria[61]. Although we did identify pathologies associated with the chronic infection model, these basal pathologies in C3H/HeJ mice may lead to an inability to identify other potential pulmonary damage associated with infection. Finally, we noted no incidence of airway smooth muscle hypertrophy, airway squamous epithelium metaplasia, airway goblet cell hyperplasia, and fibrosis in infected or mock-infected mice. In all, these results demonstrate that the chronic infection model results in significant alveolitis and airway attenuation, indicative of chronic infection.

## InvL is a critical virulence factor for long-term infection

The acute pulmonary infection model has been widely used to characterize *A. baumannii* virulence factors[21]. While the acute model is valuable for identifying bacterial proteins required at early timepoints, these mice clear infection within 3-4 dpi, not allowing for the identification of factors required for prolonged infection. As a proof of principle, we sought to determine if the chronic respiratory infection model could identify proteins required for bacterial persistence in the lungs. We hypothesized that prolonged adherence to respiratory epithelium would be required for persistence, so we first tested individual mutants lacking previously identified *A. baumannii* adhesins (Bap, Ata, FhaBC, and InvL) for attenuation in the chronic infection model (Fig. S8)[62–71]. *tlr4* mutant mice were infected with 10⁵ G636 WT or mutant bacteria, and mice were sacrificed at 1 or 14 dpi for lung colony-forming units (CFU) quantification. This experiment indicated a possible role for InvL in long-term infection as mice began to clear bacteria in the lungs by 14 dpi. We also performed more extensive analyses with G636 WT, *invL* mutant (Δ*invL*), and complemented *invL* mutant (*invL*⁺) strains in the chronic infection model, sacrificing mice at 1, 7, 14, and 21 dpi to quantify CFU in the lungs (Fig. 4A–D). At 1 dpi, no defect was

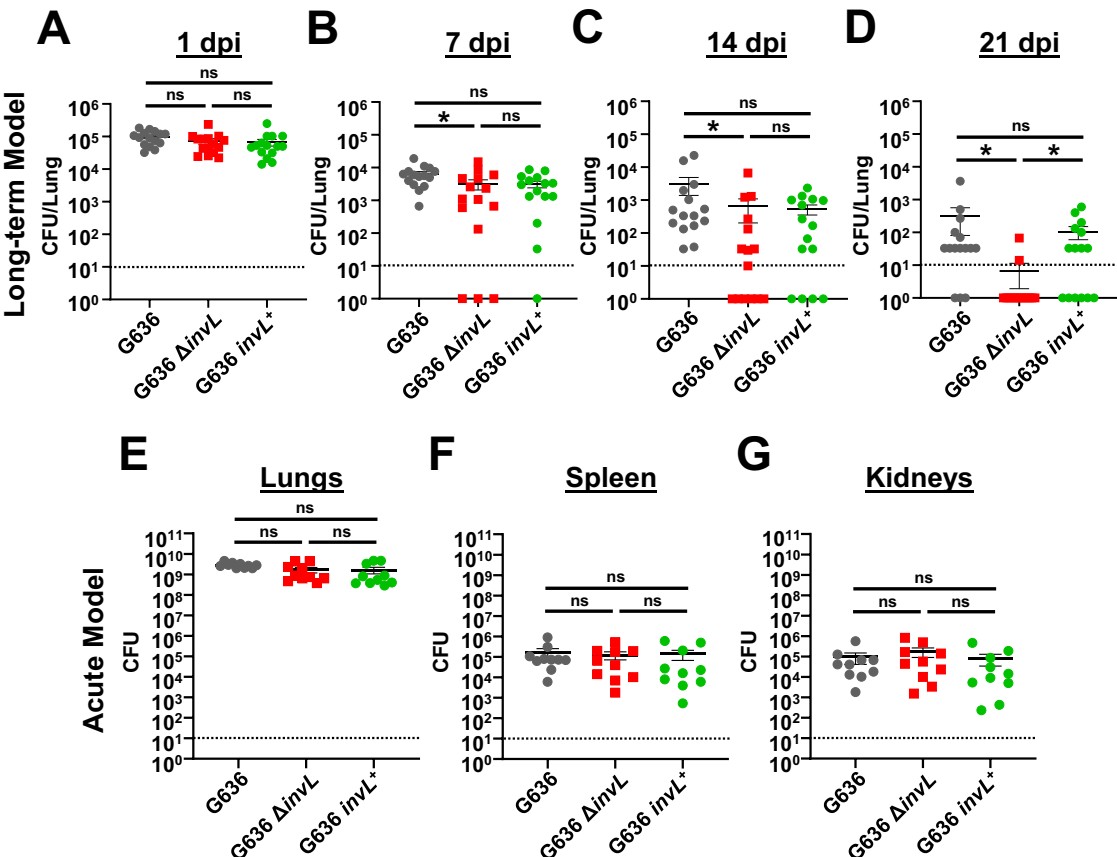

**Fig. 4 | InvL is a critical virulence factor for long-term respiratory infection, but dispensable in the acute infection model.** Female C3H/HeJ (*tlr4* mutant) mice were infected with $10^5$ G636, G636 Δ*invL*, or G636 *invL*⁺. Groups of mice were then sacrificed at 1 dpi (**A**), 7 dpi (**B**), 14 dpi (**C**), and 21 dpi (**D**), and CFU in the lungs were quantified. Shown are the pooled results from three independent experiments. For the acute infection model, groups of female C57BL/6 mice were infected with $10^9$ G636, G636 Δ*invL*, or G636 *invL*⁺. 24 hpi, mice were sacrificed, and CFU in the lungs (**E**), spleen (**F**), and kidneys (**G**) were enumerated. Shown are the pooled results of two independent experiments. Each data point represents an individual mouse, the horizontal line represents the mean, and the SEM is indicated by error bars. The limit of detection (10 CFU) is indicated by the dashed line. *$p < 0.05$; Kruskal–Wallis *H* test with Dunn's test for multiple comparisons; ns not significant. Source data and statistical test details are provided in the Source Data file.

identified between WT and mutant-infected mice. However, by 7 dpi, mutant-infected mice had significantly reduced CFU. This significant reduction in CFU was also observed at 14 and 21 dpi. At 7 and 14 dpi, genetic complementation partially rescued the phenotype, as no significant difference was detected between WT and complemented strains, although the complemented mutant did not have significantly higher CFU than the mutant strain. On day 21, full genetic complementation was observed. As high variability within groups was observed during long-term infection, we also examined each of the three pooled experiments from Fig. 4 using WT, Δ*invL*, *invL*⁺ strains individually (Fig. S9) to assess if results were consistent among biological replicates. Indeed, early during infection in these 3 independent experiments, minimal differences were seen with the Δ*invL* mutant strain relative to the WT strain. By 21 dpi, all mice in 2/3 groups infected with Δ*invL* mutant had cleared infection. Alternatively, all groups infected with WT and *invL*⁺ strains had mice still infected at that timepoint. In all, these results indicate that InvL is specifically important for long-term infection in the chronic model.

We next compared results from the chronic respiratory infection model to the acute infection model. We infected C57BL/6 mice with $10^9$ G636 WT, Δ*invL*, or *invL*⁺ strains. 24 hpi, mice were sacrificed, and CFU in the lungs, spleens, and kidneys were quantified (Fig. 4E–G). As opposed to results seen in the chronic infection model, the Δ*invL* mutant had no significant defect in bacterial load in the lungs. Additionally, no defect was noted in dissemination to the spleen and kidneys, indicating that InvL is dispensable in the acute infection model. In

all, these results highlight the differences in required bacterial genes between these disparate pulmonary infection models and show the importance of continuing to explore models that can better approximate clinical disease. Additionally, these experiments establish InvL as the first known *A. baumannii* virulence factor required for long-term infection.

## The chronic infection model can be used to study the outcome of antibiotic treatment

The acute pulmonary infection model has been employed extensively to assess effects of antibiotic treatment[21]. However, this model only allows us to estimate the efficacy of antibiotics by measuring the initial reduction in the bacterial burden at 24–36 hpi due to rapid bacterial clearance by the host. A clear limitation of this model is that it does not inform if bacterial infection is cleared, or if persistent bacteria remain in the lung. The chronic respiratory infection model therefore represents a novel platform that could be used to track the kinetics of *A. baumannii* clearance due to antibiotic treatment. As a proof of principle, we assessed the effect of tigecycline, colistin, and imipenem in the chronic model with strains G636 and G654 at antibiotic concentrations similar to those previously used to determine treatment efficacy in mice (Figs. 5A, B and S10A–D)[72–78]. We additionally assessed the effect of apramycin, a drug with demonstrated efficacy and safety in mice that is currently in Phase I clinical trials for use in humans (Fig. 5C, D)[79–81]. Minimum inhibitory concentrations (MICs) for G636 and G654 for these and other commonly used antibiotics are listed in

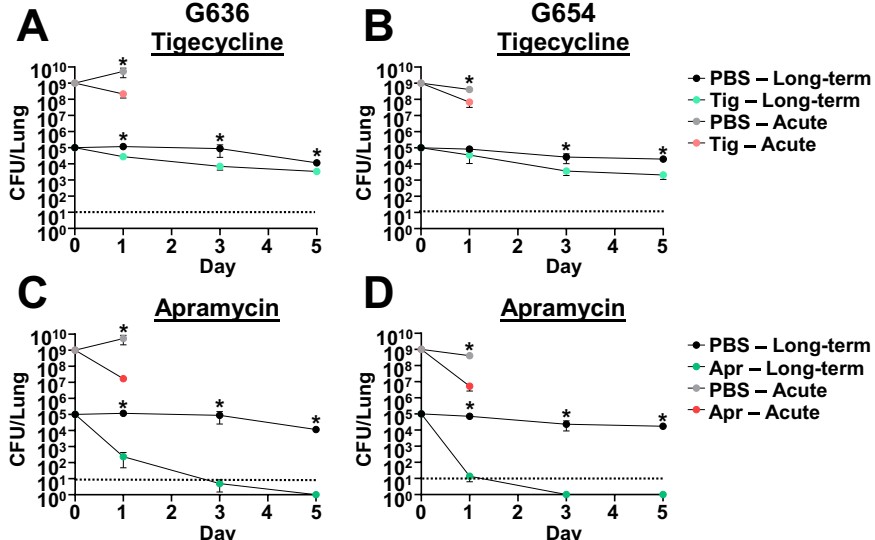

**Fig. 5 | The chronic respiratory infection model can be used to study outcomes of antibiotic treatment.** Groups of female C3H/HeJ (*tlr4* mutant) mice were infected with $10^5$ G636 (**A**, **C**) or $10^5$ G654 (**B**, **D**) and sacrificed at 1, 3, and 5 dpi (long-term). Additionally, groups of female C57Bl/6 mice were infected with $10^9$ G636 (**A**, **C**) or $10^9$ G654 (**B**, **D**) and sacrificed at 24 hpi (acute). Mice in both infection models were treated intraperitoneally with PBS or 100 mg/kg tigecycline (tig) every 12 h (**A**, **B**) or PBS or 500 mg/kg apramycin (apr) every 12 h (**C**, **D**) with all treatments beginning at 4 hpi. At each timepoint, CFU were quantified in the lungs. Shown are the pooled results from two independent experiments, each data point represents the mean, and the SEM is represented by error bars. The limit of detection (10 CFU) is indicated by the dashed line. *$p < 0.05$; two-tailed Mann–Whitney $U$ test. Source data and statistical test details are provided in the Source Data file.

Table S3. Colistin was ineffective for both strains in the acute infection model, while initial reductions in CFU were noted in the chronic infection model (Fig. S5A, B). However, bacterial numbers appeared to stabilize over time in the chronic infection model, consistent with the development of bacterial persisters (discussed below). Imipenem showed limited efficacy against both strains in both models (Fig. S10C, D), as expected given the strains' resistance in vitro (Table S3). At 24 hpi, tigecycline and apramycin treatment resulted in initial reductions in CFU in both the chronic and the acute infection models relative to PBS-treated mice (Fig. 5A–D). However, the chronic model enabled us to differentiate the efficacy of both antibiotics at later times. Apramycin treatment ultimately led to clearance after 3–5 days, demonstrating the efficacy of this antibiotic. However, with tigecycline treatment, although there were initial reductions in CFU, bacterial numbers leveled out over time indicative of treatment failure. The behavior of bacteria in presence of tigecycline over time is consistent with the development of persisters. Notably, the efficacy of tigecycline and apramycin against *A. baumannii* cannot be distinguished at 24 hpi. These results indicate that the chronic model can be used to determine outcome of infection with therapeutic intervention, a significant advantage over the currently employed acute infection model.

**Use of the chronic infection model to study bacterial co-infections reveals that *Staphylococcus aureus* exacerbates ongoing *A. baumannii* infection while *Klebsiella pneumoniae* leads to earlier clearance**

Approximately 25% of *A. baumannii* pulmonary infections are polymicrobial, and two of the most commonly co-infecting pathogens are *Staphylococcus aureus* and *Klebsiella pneumoniae*[9]. We thus sought to assess the impact of secondary infections with these two bacteria on the outcome of *A. baumannii* infection in the context of the chronic respiratory infection model. For these experiments, we first established a primary *A. baumannii* infection by inoculating *tlr4* mutant mice with $10^5$ CFU of strain G636. Following 14 days of *A. baumannii* infection, we inoculated mice with $5 \times 10^7$ CFU of *S. aureus* strain Newman or *K. pneumoniae* strain TOP52, mock-treated mice with PBS, or left mice untreated. One and 2 days post-secondary infection, mice were sacrificed, and bacterial CFU were quantified in the lungs,

spleens, and kidneys (Figs. 6 and S11, S12). Secondary infection with *S. aureus* led to a resurgence of *A. baumannii* CFU in the lungs of many mice, though the overall mean CFU in these mice were not significantly different from mock-infected and untreated groups (Fig. 6A, D, E). *A. baumannii* were also identified in the spleens and kidneys of some mice that received the secondary *S. aureus* infection, even though *A. baumannii* bacteremia rarely occurs in the context of this chronic respiratory infection model (Fig. S11). Additionally, *S. aureus* trended toward increased numbers in the lungs, spleens, and kidneys in the context of polymicrobial infection with *A. baumannii* relative to monomicrobial infection (Fig. S12A–C). Contrarily, secondary infection with *K. pneumoniae* significantly decreased *A. baumannii* CFU in the lungs relative to mock-infected and untreated groups at 1 dpi (Fig. 6A, D, E). Additionally, polymicrobial infection with *A. baumannii* and *K. pneumoniae* resulted in significantly reduced *K. pneumoniae* CFU recovered in the lungs at 1 and 2 dpi, as well as in the spleens and kidneys at 2 dpi relative to *K. pneumoniae* monomicrobial infection (Fig. S7D–F). Although understanding the interactions between these bacteria is beyond the scope of this work, these experiments indicate that *S. aureus* exacerbates *A. baumannii* infection, while *K. pneumoniae* attenuates infection in the context of the chronic infection model. Additionally, these results demonstrate the ability of the model to be used to study longer-term aspects of polymicrobial interactions that were not previously able to be done with the acute infection model.

## Discussion

*A. baumannii* has emerged as a significant cause of nosocomial pneumonia and is of major clinical importance due to its extremely high rates of multidrug resistance[8,12,13]. Despite this, our understanding of *A. baumannii* respiratory pathogenesis is hindered by a shortage of clinically relevant infection models. Here, we aimed to address this significant gap in the field by developing a novel respiratory infection model. In this pursuit, we found that, at likely more clinically-relevant inoculums, *tlr4* mutant mice maintain long-term respiratory infections by *A. baumannii*. We then demonstrate the versatility of this model which enabled (1) the identification of a bacterial virulence factor required for long-term respiratory infection, which is not required in acute models, (2) the study of kinetics of bacterial clearance upon

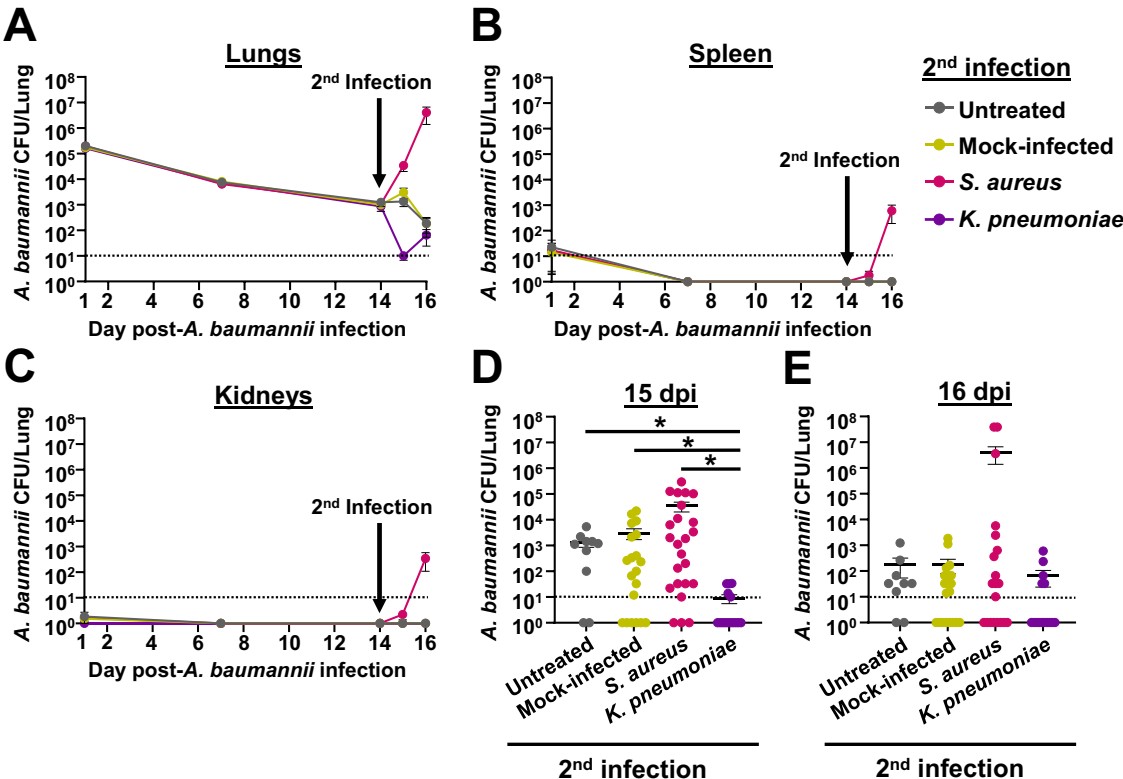

**Fig. 6 | Bacterial secondary infection alters the course of chronic *A. baumannii* pneumonia.** Female C3H/HeJ (*tlr4* mutant) mice were intranasally inoculated with 10^5 G636, and groups of mice were sacrificed at 1, 7, and 14 dpi. At 14 days post-*A. baumannii* infection, groups of mice were either not inoculated (untreated), inoculated with PBS (mock-infected), infected with *S. aureus*, or infected with *K. pneumoniae*. Subsequently, on days 15 and 16 post-*A. baumannii* infection (1 and 2 days post-secondary infection), mice were sacrificed, and *A. baumannii* CFU were quantified in the lungs (**A**, **D**, **E**), spleen (**B**), and kidneys (**C**). In **A**–**C**, each data point represents the mean, the SEM is represented by error bars, and the limit of detection (10 CFU) is indicated by the dashed line. In **D** and **E**, each data point represents an individual mouse, the horizontal line represents the mean, and the SEM is indicated by error bars. Shown are the pooled results from at least two independent experiments. *$p < 0.05$; Kruskal–Wallis *H* test with Dunn's test for multiple comparisons. The limit of detection (10 CFU) is indicated by the dashed line. Source data and statistical test details are provided in the Source Data file.

treatment with clinically-relevant antibiotics, and (3) the exploration of the impact of secondary infections with two commonly co-isolated respiratory pathogens.

Importantly, use of a long-term immunocompromised mouse model in this manuscript maintains clinical relevance, as *A. baumannii* primarily affects immunocompromised and critically ill patients, causing extended infections particularly in patients in intensive care units (ICUs)[7,8,10,11]. Furthermore, studies have indicated a significant link between human *tlr4* mutations and poor clinical outcomes from *A. baumannii* infection, further supporting the use of a *tlr4* mutant murine model to study pathogenesis[39–41]. For example, Behairy et al. demonstrated that *tlr4* polymorphisms were associated with significantly increased *A. baumannii* infections, but not infections with other Gram-negative pathogens, in Egyptian ICUs[39]. Similarly, Chatzi et al. showed that *tlr4* polymorphisms were associated with increased risk of *A. baumannii* infections in Greek ICUs[40]. Finally, using the large study population of the Chinese National Hospital Infection Surveillance Network, He et al. demonstrated a significant link between *tlr4* mutations and *A. baumannii* infection[41].

While use of *tlr4* mutant mice enables long term *A. baumannii* pulmonary infection and has clear relevance to human disease, it is important to note that the chronic infection model described here has limitations. First, it is not possible for one model to reflect the state of all populations that are susceptible to *A. baumannii* infection and all associated diseases/outcomes; *A. baumannii* is capable of infecting individuals with a range of immunocompromisation and infection duration is often the result of multi-drug resistance in combination

with immunocompromised status[82]. Moreover, *A. baumannii* can cause a range of disease from mild to severe pneumonia depending on the individual[83]. Therefore, careful consideration should be taken when choosing a murine model, especially in the case of studying host responses which will drastically differ depending on if immunosuppression is used and the mechanism of immunosuppression. Related to this limitation, an important aspect of this model that likely enabled bacterial persistence was the decreased immune response associated with low inoculums in *tlr4* mutant mice. In the case of studying exacerbated/lethal host responses that can occur in severe human infections and pathologies associated with specific immune mechanisms, future studies may need to employ mice lacking specific immune factors of interest rather than TLR4. For these circumstances, the previously described acute infection model using C57BL/6 mice and congenic mutants will likely still serve as a valuable tool[21]. Second, use of a *tlr4* mutant murine model may impact potential bacterial virulence mechanisms that are able to be studied. For example, when studying bacterial mutations potentially altering bacterial LOS, a *tlr4* mutant mouse model may have limited use. A third limitation of this model is that its use may be strain-dependent. While we established that 3 modern respiratory isolates and one lab-domesticated strain, *Ab*17978, establish long-term infection, another lab-domesticated strain, *Ab*19606, was cleared by 24 hpi. Therefore, strain-to-strain variability could affect the ability to use the model, and preliminary experiments may be required to adapt the model to strains of interest. While the chronic infection model does have some limitations, similar to all murine models, it is the first model enabling the study of *A. baumannii*

persistence past 1 week pi. It therefore has the potential to change how we address questions associated with specifically long-term infection, which is extremely relevant in the case of difficult to treat multidrug resistant infections due to treatment failures. Future work, however, will be aimed at identifying potential other methods of extending infection in mice that can be used in conjunction with the chronic infection model reported here. Such alternatives include the use of aged mouse models (klotho mutation) or mixing bacteria with gastric mucin, both of which have been shown to extend infection kinetics in sepsis models[84–89].

In this study, we found that InvL is required for chronic infection, and, more importantly, at the later stages of infection. However, InvL was dispensable in the context of the acute infection model. There are multiple possible reasons for this discrepancy. First, the massive bacterial dose required for the acute infection model may mask potential defects that can now be detected with a smaller, more clinically-relevant inoculum. This is unlikely, as WT and *invL* mutant bacteria behave similarly at early time points in our model. An alternative reason could be that adhesins required early during infection/interaction with the healthy airway differ from those required during persistent interaction with a more inflamed or damaged airway. It is well-established that the airway extracellular matrix (ECM) is altered by bacterial infection, lung damage, and/or inflammation[90,91]. Long-term lung damage and inflammation results in increased fibronectin, collagen, laminin, and fibrinogen in the ECM[92–97]. Moreover, specific pathogens elicit different inflammatory responses, resulting in distinct changes to the lung ECM. For example, in an acute mouse model of pneumonia, *Pseudomonas aeruginosa* induces versican deposition in the lungs, while *Escherichia coli* induces robust versican and hyaluronan deposition[98,99]. We previously showed that InvL can bind α5β1 integrin, collagen V, and fibrinogen[71]. However, whether *A. baumannii* infection or the associated inflammation induces production of these protein(s) during pulmonary infection is unknown. Future work will investigate this possibility, as well as assess which InvL-host protein interactions are essential for chronic infection.

Herein, we demonstrate the potential to use the chronic respiratory infection model to study the efficacy of antibiotic treatments over time. One intriguing finding from these experiments is that with antibiotics such as colistin and tigecycline, an initial decrease in CFU (~10–100 fold) recovered from the lungs at 1 dpi was observed. However, following this decrease, the number of bacteria in the lungs appeared to stabilize over time. It is tempting to speculate that this is the result of the formation of bacterial persisters, defined as bacterial cells that become tolerant to antibiotics despite undergoing no genetic changes[100–102]. Importantly, the commonly used acute infection model does not allow for the study of bacterial persisters due to the short time course of the model. Given that persister cells represent a major cause of treatment failure and chronic infection, the chronic infection model presented here represents a unique platform that is desperately needed to understand this aspect of *A. baumannii* pathogenesis. Furthermore, our model offers new possibilities to study efficacy of novel antibiotics in murine models before committing to expensive clinical trials.

While a significant portion of *A. baumannii* infections are polymicrobial, the acute infection model has limitations for use with polymicrobial infections. First the quick clearance of the bacteria usually only allows inoculation at a single timepoint, thus not enabling investigation of secondary infections. Second, the high required infectious dose often means that typical inoculums for bacteria used in these experiments must be adjusted, so mice do not succumb to infections at early timepoints. Here, we applied the chronic respiratory infection model to assess the result of secondary infection with two pathogens commonly co-isolated with *A. baumannii*, *S. aureus* and *K. pneumoniae*. We found opposite results with these different bacteria; *S. aureus* secondary infection trended toward exacerbation of *A.*

*baumannii* infection, while *K. pneumoniae* secondary infection led to reduced *A. baumannii* numbers. The potential synergism of *A. baumannii* and *S. aureus* in the chronic infection model aligns with previous reports. For example, using a Tn-Seq-based approach, Li et al. demonstrated that the 49% of genes required by *S. aureus* for monomicrobial infection in a murine systemic infection model became non-essential upon *A. baumannii* co-infection[103]. Another recent report showed that *S. aureus* can support *A. baumannii* growth in vitro by providing acetoin as a carbon source[104].

Although we found that *K. pneumoniae* secondary infection led to reduced *A. baumannii* numbers in the lungs in the chronic infection model, one study has shown that *K. pneumoniae* could cross-feed *A. baumannii* through products of sugar fermentation in vitro and demonstrated that co-infection led to reduced survival of *Galleria mellonella* relative to monomicrobial infection with either pathogen[105]. This, in part, shows that these two pathogens can have beneficial interactions for the bacteria. There are two potential reasons however for the reduction of CFU for both bacteria in the context of the chronic infection model reported here; (1) bacterial competition or (2) the host response to the secondary infection. Regarding bacterial competition, there have been several lines of evidence pointing to direct bacterial killing between diverse *A. baumannii* and *K. pneumoniae* strains mediated by the type VI secretion system[106–109]. In addition to direct killing, this bacterial competition could be indirect as well, as both *A. baumannii* and *K. pneumoniae* may be competing for similar nutrients in the lung microenvironment. With respect to the immune response, a difference between this work and the above study is that the microenvironment encountered in the mammalian lung is not perfectly modeled by the wax moth[110]. Our results may therefore be the result of TLR4-independent host response elicited by the combination of both bacteria that is not recapitulated by a *G. mellonella* model. While understanding the precise mechanism behind the in vivo interactions between *A. baumannii* and commonly co-isolated pathogens is outside the scope of the current study, these results highlight the practicality of applying the chronic respiratory infection model to better understand polymicrobial infections.

In this study, we have validated several different uses for the chronic respiratory infection model. However, there are also other potential uses for this model that were not previously investigable. For example, we can now perform experiments differentiating between virulence factors required for establishment of infection and factors required for maintenance of infection, assessing bacterial evolution during long-term infection, investigating changes in the pulmonary microbiome due to infection over time, and analyzing the long-term outcomes of novel therapies such as newly developed phage cocktails. Additionally, while this model was initially developed to study *Acinetobacter* respiratory infections, it has the potential to be applied to research with other respiratory pathogens in cases where suitable animal models are lacking. In all, this work describes the longest-term infection model available to investigate *A. baumannii* host-pathogen interactions to date, which will ultimately aid in the development of novel therapeutics to combat infection by this increasingly multidrug-resistant bacterium.

## Methods
### Ethical regulations
This research complies with all relevant ethical regulations and has been approved by the Washington University School of Medicine Institutional Animal Care and Use Committee (IACUC; protocol number: 23-0071) and Institutional Biosafety Committee (IBC; protocol number: 12098 Ver.25).

### Bacterial plasmids, strains, and growth conditions
Plasmids and strains used in this study are detailed in Table S4. Bacterial cultures were grown at 37 °C in Lennox broth/agar supplemented with 10 µg/mL chloramphenicol, 50 µg/mL apramycin, 100 µg/mL

ampicillin, 50 µg/mL kanamycin, 10 µg/mL tetracycline, or 10% sucrose when appropriate. All strains used in this manuscript will be made available upon request.

## Murine pneumonia models

All animal experiments were approved by the Washington University Animal Care and Use Committee, and we have complied with all relevant ethical regulations. For all rodent housing rooms, the light cycle is kept at 12 h:12 h light:dark, with lights on at 6 a.m. and lights off at 6 p.m. Temperature range is kept between 20–24°C, and humidity range is 30–70%. The acute pneumonia model was performed similar to previously described experiments[21,111]. Briefly, overnight cultures were subcultured at a 1:200 dilution and grown shaking at 37 °C for 3 h to mid-exponential growth phase. Six- to eight-week-old female C57BL/6 mice (Charles River Laboratories, Wilmington, MA) anesthetized with 4% isoflurane were intranasally inoculated with approximately $10^9$ CFU that were twice-washed in PBS. At 24 hpi, mice were sacrificed, and CFU in the lungs, spleen, and kidneys were quantified by serial dilution plating the homogenized organs. For experiments with female C3H/HeN (Envigo International Holdings, Indianapolis, IN) and C3H/HeJ (Jackson Laboratory, Bar Harbor, ME) mice, *A. baumannii*, *S. aureus*, and *K. pneumoniae* inoculums were prepared and mice were intranasally inoculated as described above, with the exception that inoculums of approximately $10^3$, $10^4$, $10^5$, $10^6$, $10^7$, or $10^8$ CFU were used for *A. baumannii*, and $5 \times 10^7$ CFU was used for *S. aureus* and *K. pneumoniae*. Following, at the indicated timepoints, mice were sacrificed and bacteria in the lungs, spleen, and kidneys were quantified as described above. For co-infections, *A. baumannii* was distinguished from *S. aureus* and *K. pneumoniae* by plating on LB agar supplemented with 10 µg/mL chloramphenicol. For antibiotic treatment experiments the indicated mice were treated intraperitoneally with PBS or 100 mg/kg tigecycline every 12 h, PBS or 5 mg/kg colistin every 8 h, PBS or 500 mg/kg apramycin every 12 h, or PBS or 100 mg/kg imipenem every 12 h with all treatments beginning 4 hpi. Antibiotics for intraperitoneal treatments were dissolved in PBS, and the injection volume was 100 µl. For cyclophosphamide treatments, one treatment of 300 mg/kg dissolved in PBS was intraperitoneally administered 3 days prior to infection.

## Flow cytometry

Flow cytometry was performed similarly to previously described methods[43]. Briefly, BALF samples were collected in PBS supplemented with 1 mM EDTA, and cells were collected by centrifugation at $300 \times g$ for 5 min. Cells were then resuspended in Pharm Lyse Buffer (BD Biosciences, Franklin Lakes, NJ) and incubated for 3 min at room temperature to lyse red blood cells. Cells were subsequently washed in fluorescence-activated cell sorting (FACS) buffer (PBS supplemented with 1% heat inactivated fetal bovine serum and 0.1% sodium azide) and blocked with TruStain FcX PLUS (BioLegend, San Diego, CA; Ref: 156603) for 15 min at 4 °C. Samples were then stained with a 1:100 dilution with anti-CD45-BV605 (BioLegend; Ref: 103139), anti-CD11b-Alexa700 (BioLegend; Ref: 101222), anti-CD11c-APC (BioLegend; Ref: 117309), anti-SiglecF-PerCP5.5 (BioLegend: Ref: 155525), and anti-Ly6G-BV421 (BioLegend; Ref: 127628) for 30 min at 4 °C, markers used to define total AMs and PMNs. In some experiments, CD45 antibody was not included in the staining panel. Following, cells were washed in FACS buffer and fixed in 2% paraformaldehyde (PFA). Samples were read on a LSR II Fortessa cytometer (BD Biosciences) or an Aurora cytometer (Cytek Biosciences, Fremont, CA). Total cell counts in the BALF were calculated using Precision Count Beads (BioLegend) according to the manufacturer's instructions.

## Histopathology of lung slices

Lungs were perfused with in PBS followed by either inflation with 10% neutral buffered formalin and paraffin embedding or inflation with

optimal cutting temperature (OCT) compound (Fisher Scientific) diluted 1:1 in 4% PFA followed by snap freezing. Lungs were sectioned at 4–5 µm thickness, and staining with hematoxylin and eosin (H&E). Histopathological evaluation and scoring were performed by a board-certified veterinary pathologist, and lungs were assessed for changes in alveolitis, alveolar histiocytosis, perivascular infiltrate, peribronchiolitis, BALT, loss and attenuation of airway epithelium, pleuritis, airway smooth muscle hypertrophy, airway squamous epithelium metaplasia, airway goblet cell hyperplasia, and fibrosis.

## Generation of constructs and strains used in this study

Primers used in this study are listed in Table S5. DNA fragments were assembled using either the In-Fusion HD EcoDry Cloning Kit (TaKaRa Bio, Mountain View, CA) or NEBuilder HiFi DNA Assembly Master Mix (New England Biolabs, Ipswich, MA). To generate the vector for generation of the *invL* mutational construct, pEX18Tc was amplified without the tetracycline resistance cassette (primers: 5′ pEX18 marker swap and 3′ pEX18 marker swap), the apramycin resistance cassette was amplified from pKD4-Apr (primers: 5′ Apr for pEX18Ap and 3′ Apr for pEX18Ap), and the amplicons were assembled, generating pEX18Ap[112,113]. The pEX18Ap mutational constructs were then made by amplifying the pEX18Ap vector (primers: 5′ pEX18Tc and 3′ pEX18Tc), a -1000 bp region upstream of the genes of interest (*invL*KO primers: 5′ F1 G636 *invL*KO and 3′ F1 G636 *invL*KO; *bap*KO primers: 5′ F1 G636 *bap*KO and 3′ F1 G636 *bap*KO; *ata*KO primers: 5′ F1 G636 *ata*KO and 3′ F1 G636 *ata*KO; *fhaBC*KO primers: 5′ F1 G636 *fhaBC*KO and 3′ F1 G636 *fhaBC*KO), and a -1000 bp region downstream of the genes of interest (*invL*KO primers: 5′ F2 G636 *invL*KO and 3′ F2 G636 *invL*KO; *bap*KO primers: 5′ F2 G636 *bap*KO and 3′ F2 G636 *bap*KO; *ata*KO primers: 5′ F2 G636 *ata*KO and 3′ F2 G636 *ata*KO; *fhaBC*KO primers: 5′ F2 G636 *fhaBC*KO and 3′ F2 G636 *fhaBC*KO), followed by assembly of these amplicons. Mutational constructs were then transformed into G636, and strains with the integrated plasmid were selected for by apramycin treatment. Counterselection for double crossover was performed by plating these strains on LB agar without NaCl supplemented with 10% sucrose. Mutants were then confirmed by PCR analyses and whole-genome sequencing.

The *invL* complementation construct was generated by amplifying the putative promoter region (-300 bp upstream) along with the *invL* open reading frame (primers: 5′ G636 *fdeC*KO Comp and 3′ G636 *fdeC*KO Comp-His6 v2) and the pUC18T-miniTn7T-Apr vector (primers: Tn7 linear Fwd-His6 and Tn7 liner Rev)[114]. These amplicons were then assembled, generating pUC18T-miniTn7T-Apr::G636 *invL*KO comp. To generate the *gfp* integration construct, the *gfp* cassette was amplified from PB-FLuc+GFPd2 (primers: 5′ d2EGFP for pUC18T-mTn7 and 3′ d2EGFP for pUC18T-mTn7) and pUC18T-mTn7-Apr was amplified (primers: 5′ pUC18T-mTn7 for d2EGFP and 3′ pUC18T-mTn7 for d2EGFP). These fragments were then assembled, generating pUC18T-miniTn7T-Apr::*gfpd2*. pUC18T-miniTn7T-Apr::G636 *invL*KO comp and pUC18T-miniTn7T-Apr::*gfpd2* were introduced into G636 Δ*invL* and G636, respectively, using a four-parental conjugation technique, as previously described[114–117]. Selection was achieved using LB supplemented with apramycin and chloramphenicol, and insertion of the respective fragments at the mTn7 site in the resulting G636 *invL*+ and G636-*gfp* strains was confirmed by PCR analyses.

## Statistical methods

All statistical analyses were performed using GraphPad Prism version 9, and $p$ values < 0.05 were considered statistically significant. The two-tailed Mann–Whitney $U$ test was used when comparing two groups, and the Kruskal–Wallis $H$ test with Dunn's test for multiple comparisons was used when comparing more than two groups. Two-way ANOVA with Tukey's or Bonferroni's test for multiple comparisons was used when comparing two or more groups with two independent variables. Source data are provided as a source data file.

**Reporting summary**

Further information on research design is available in the Nature Portfolio Reporting Summary linked to this article.

## Data availability

The genome sequence for *A. baumannii* strain G636 has been deposited into NCBI (Accession: PRJNA1182817). All data in the publication are presented in figures and all values are available as a Source Data file. Source data are provided with this paper.

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

## Acknowledgements

This work was supported by funding to M.F.F. (R01AI166359), C.J.L. (T32AI007172), and C.B.L. (R01AI137062) through the National Institute of Allergy and Infectious Diseases of the National Institutes of Health. J.M. was supported through The American Association of Immunologists Careers in Immunology Fellowship Program and The Pediatric Cardiovascular and Pulmonary Research Training Program (5T32HL125241-07). The modern respiratory isolates, G636 (strain 3689) and G654 (strain 6919), were collected by the CDC-funded Georgia Emerging Infections Program's (EIP) Multi-site Gram-Negative Surveillance Initiative (MuGSI) and kindly provided by Sarah Satola. The modern respiratory isolate, G803 (Strain ABM452), was provided by Masoumeh Douraghi of the Division of Microbiology in the Department of Pathobiology at Tehran University of Medical Sciences School of Public Health. We also acknowledge Jennifer Philips, Jacco Boon, and Gayan Bamunuarachchi for thoughtful discussion about the manuscript. Flow cytometry work was supported by Asya Smirnov and the Center for Women's Infectious Disease Research and Department of Molecular Microbiology Flow Cytometry Facility at Washington University School of Medicine. We thank Wandy Beatty and the Washington University School of Medicine Molecular Microbiology Imaging Facility for microscopy assistance, staff in the Washington University School of Medicine Department of Comparative Medicine Research Animal Diagnostic Laboratory for histology slide preparation, and Dakota Hall for technical assistance with experiments.

## Author contributions

C.J., G.D., and M.F. conceptualized the project. All authors (C.J., G.D., M.J., I.C., J.M., L.W., D.R., C.L. and M.F.) contributed to experimental design. C.J., G.D., M.J., I.C., J.M. and L.W. performed all experiments. C.J. wrote the manuscript and the remaining authors (G.D., M.J., I.C., J.M., L.W., D.R., C.L. and M.F.) edited the document.

## Competing interests

The authors declare no competing interests.
