## [Transparent Peer Review file · Nature Communications]

A chronic murine model of pulmonary *Acinetobacter baumannii* infection enabling the investigation of late virulence factors, long-term antibiotic treatments, and polymicrobial infections

Corresponding Author: Dr Mario Feldman

Version 0:

Reviewer comments:

Reviewer #1

(Remarks to the Author)

This paper describes the development of a long-term *Acinetobacter baumannii* mouse model (~3 week infections) using two recent *A. baumannii* clinical isolates. Currently the majority of *A. baumannii* mouse disease models last only a few days, require high doses and are often using lab-adapted strains. The model described here is interesting as it utilises lower doses than many other *A. baumannii* mouse models and gives long-term bacterial colonisation. Utilisation of the new model allowed identification of a virulence factor that appears to be specific for long-term carriage. Furthermore, the model was used to assess polymicrobial interactions and looks useful for testing efficacy of antibiotics over longer time periods. The work will be of interest to those that work with *A. baumannii* and may be relevant to development of mouse models for some other pathogens, although that is not shown in this work.

Major Points

1. How phylogenetically similar are the two clinical strains that were tested (G636 and G654). Are they diverse enough to give insight into general applicability?
2. A number of times it is noted that high dose models are likely not clinically relevant, although the natural clinically relevant dose is difficult to predict and may fluctuate widely, especially where infections may follow direct inoculation of indwelling devices that may be contaminated with a biofilm. It initially appears surprising that in *tlr4* mutant mice the high dose is cleared more rapidly than the low dose (e.g high doses mostly cleared by 7-10 days and low doses not often cleared even after 19 days). It would be important to show a wider range of doses to compare when this long-term survival becomes most marked and when, or if, the *tlr4* mice can completely clear the infection. Specifically, would a lower dose of 10^3 give increased, reduced or similar levels of organisms over the 19 days? Is 10^5 the optimal dose in this model? Understanding how sensitive the model is to dose will guide more widespread and appropriate uptake.
3. For the detection of intracellular G636, did the GFP-labelled strain grow at the same rate as the WT G636 strain?
4. Are there only 2 mice for the G654 initial WT infections (Fig. 1)?
5. Fig 2. In panel A. Is 10^5 vs 10^8 for TLR4 mutant not significant? Please check.
6. The only significant difference in cytokines between low dose infections in WT and low dose infections in *tlr4* mice was GM-CSF. It would be very interesting to test the outcome of blocking the action of this cytokine in WT mice.
7. Fig. S2 has panel B in the image labelled 2 h pi when it is 2 day pi
8. In figure S4 are differences for *invL* mutant significant at 14 days?
9. Fig. 4, panel G not labelled correctly (currently E). Also, the legend is incorrect here too, as panels E, F and G are listed as A, B and C.
10. It is claimed that complementation rescued the *invL* complemented strain. However, comparison of the complemented mutant against the mutant showed no statistically significant change.
11. Is there any evidence that *InvL* antibodies are observed in long-term human infections?
12. In Figure 5 each data point is labelled as representing an individual mouse, but only single dots are visible. Furthermore, the horizontal line is not really a mean as it connects the means from each time point but actually is mostly interpolated data.
13. Figure 6. For panels A, B and C, please label the starting x axis value (1 not 0). Also, there is meant to be a limit of detection dotted line, but I can't see it. Is it 1 CFU so lost in the base line?

14. In the description of Figure 6 on lines 276, it states that *K. pneumoniae* significantly decreased *A. baumannii* CFU in the lungs relative to mock infected and untreated groups; however, this only appears to be significant at 1 day p.i. and not 2 days p.i.
15. It is speculated that one reason for the reduced survival of *A. baumannii* and *K. pneumoniae* in polymicrobial infections may be the presence of a T6SS. This should be tested directly with an *A. baumannii* tssM mutant.
16. I think it would be helpful to have some comments/discussion on,
 - a. Why a tlr4 knockout model might be a more relevant model than wt mice when both humans and mice have tlr4. Do you expect the model would still have relevance for analysis of bacterial mutants with altered LOS?
 - b. 7-day model using Klotho mice (longer term model, higher dose, older strain).
 - c. Low dose infections together with gastric mucin (10^5 CFU, acute model, with older strain, non-respiratory).

Minor Points

1. Fig. 1 legend "Each data point indicates and individual" should read "Each data point indicates an individual"
2. Fig. S1 legend "Following, bacterial CFU" should read "Bacterial CFU"
3. Figure 5 legend, "were treated intraperitoneally treated with PBS" should read "were treated intraperitoneally with PBS"
4. Figure 6 legend, "were infected on *K. pneumoniae*" should read "were infected with *K. pneumoniae*"
5. Line 280 should refer to Fig. S7D-F not just D-E

Reviewer #2

(Remarks to the Author)

In their manuscript, Dr. Jackson-Litteken and colleagues reported their efforts to develop a mouse model of chronic respiratory *Acinetobacter baumannii* (Ab) infection using tlr4 mutant mice (C3H/HeJ) and a low bacterial inoculum. Using this model, the authors investigated the potential role of adhesin InvL as the key virulence factor in establishing chronic infection. Attempts have also been made to demonstrate the potential utility of this model for evaluating antibiotic efficacy and studying polymicrobial infections.

The mouse model reported in this manuscript is new and interesting. If validated, it will complement other available mouse models for infection pathogenesis studies and potential therapeutic evaluation. However, several experiments (such as the mutant study and the polymicrobial infection) were largely descriptive and more in-depth studies will be needed. In particular, the robustness of the model and the role of InvL-mediated persistent infection have suffered from the high variations in the tissue bacterial burdens between individual mice, even though statistical significance is achieved in some instances.

1. The authors tried to demonstrate that the ability to establish the chronic infection is somewhat associated with the modern isolates by using ATCC19606 as a control. To support this assumption, additional old strains will need to be included since ATCC19606 is a low-virulence strain in mice and may not be an ideal representation of the old strains.
2. Mechanisms of chronic infection in this model: From the pulmonary cellular and cytokine response data, the authors have ruled out their contribution to the development of chronic infection, which leaves its mechanisms to be further determined. It will be interesting to determine the potential role of PMNs in the development of the chronic infection in this model.
3. Histopathology analysis (Fig. 3): Was the histologic analysis and scoring done by a laboratory animal pathologist? Some chronic histopathologic changes (such as squamous epithelium metaplasia and BALT formation etc.) were observed as early as 4 hpi. Any reasons for this?
4. The role of InvL in the persistence of Ab in the lung: The data on the mutant and complementation strain were highly variable between individual mouse. Additional in vitro cell culture and in vivo data to demonstrate the increased adhesin expression and bacterial adhesion to the respiratory epithelial cells in the chronic infection will be needed to support the overall hypothesis.
5. Please indicate the detection limits for cytokines and all the bacterial burden assays.
6. Fig. 1: The legend indicates that the samples were collected every 3 days (Ln 575), which appears to be inconsistent with the plots in the figure. Also, please correct a typo on Ln 576 (and to an).
7. As anticipated, the mice inoculated with a high dose of Ab have cleared the infection rapidly (Fig. 1). However, the number of PMNs in the lung of WT mice (Fig. 2F) remains significantly elevated at dpi 7. This observation is inconsistent with published data although the different Ab and mouse strains were used in these studies. Some discussions on this will be useful.
8. Fig. 4: A large number of mice were used in this experiment. The legend indicates that the results shown are from 3 independent experiments. Can the authors clarify if the data shown in the figure are one of the 3 independent experiments or the results of 3 pooled experiments? If the latter, it will be important to show if the result from each individual experiment supports the conclusion, given the high intragroup variation in some groups.
9. Some discussions on how this model reflects the human infection are needed since the long clinical duration of the Ab infection could largely be due to the antibiotic resistance of the Ab strain and may not be associated with a specific immune

factor such as TLR 4.

Reviewer #3

(Remarks to the Author)

In this manuscript, the authors report the development and application of a novel chronic murine model of pulmonary *Acinetobacter baumannii* infection. Current available murine pulmonary *Acinetobacter* models are all based on acute infection that require a very high bacterial inoculum (10^8 - 10^9). This artificial host-pathogen interaction might skew many aspects of *Acinetobacter* pathogenesis. The authors reported a very exciting chronic *Acinetobacter* murine model allows for studying late virulence factors, long-term antibiotic treatments, and polymicrobial infections. Overall, the study is well designed and presented. Here are some minor comments:

1. Based on the Fig. 4E-F bacterial burden data, the authors concluded that InvL is dispensable in the acute infection. Does the invL mutant also kill the wt and tlr4 mutant mice similar to G636 in the acute model? Please correct the legend describing acute infection Fig. 4E-F.
2. Please expand the discussion regarding the relevance of this novel chronic infection model to clinical disease (e.g. immune response, bacterial persistence, etc.). Also, provide insight into the potential limitation and consideration when using this long-term model, since TLR4 plays an important role in bacterial recognition and innate immunity against *A. baumannii*.
3. Please confirm the label of Figure S2 B is 2 dpi.

Version 1:

Reviewer comments:

Reviewer #1

(Remarks to the Author)

This is a review of revision one of the article by Jackson-Litteken et al.

During initial review I asked for numerous revisions. Mostly, these have been addressed adequately, and I believe the manuscript is much improved. I have noted each initial question and my interpretation of the authors responses below. I have one new point that I believe should be addressed, and this is noted after the list of questions and responses.

As noted in my original review, I believe this work will be of interest to those that work with *A. baumannii* and may be relevant to development of mouse models for some other pathogens, although that is not shown in this work. Utilisation of the new model allowed identification of a virulence factor that appears to be specific for long-term carriage. Furthermore, the model was used to assess polymicrobial interactions and looks useful for testing efficacy of antibiotics over longer time periods.

Question 1. How phylogenetically similar are the two strains that were tested (G636 and G654)?

Response: The two initial strains used were indeed highly phylogenetically similar. A new strain with a different sequence has now been included. While it would have been more comprehensive to determine whole genome sequences (rather than just MLST sequence types) and place all the strains (including the ATCC strains) on a whole genome phylogeny, the added strain addresses my main concern here.

Question 2. Why was 10^5 CFU chosen? Would a lower dose of 10^3 give increased, reduced or similar levels of organisms over the 19 days? Is there a linear response or is 10^5 the optimal dose in this model?

Response: The authors have now added in data for different starting doses, which appropriately addresses this concern.

Question 3. For the detection of intracellular G636, did the GFP-labelled strain grow at the same rate to the WT G636 strain?

Response: A full growth curve has now been added showing the two strains do grow at similar rates

Question 4. Are there only 2 mice for the G654 initial WT infections (Fig. 1)?

Response: Now clarified

Question 5. Fig 2. In panel A. Is 105 vs 108 for TLR4 mutant not significant? Please check.

Response: Was significant and changed appropriately

Question 6. The only significant difference in cytokines between low dose infections in WT and low dose infections in tlr4 mice was GM-CSF. It would be very interesting to test the outcome of blocking the action of this cytokine in WT mice.

Response: While this was not performed, I accept the suggestion that this may be beyond the scope of this manuscript

Question 7. Fig. S2 has panel B in the image labelled 2 h pi when it is 2 day pi

Response: Corrected appropriately

Question 8. In figure S4 are differences for invL mutant significant at 14 days?

Response: Clarified appropriately

Question 9. Fig. 4, panel G not labelled correctly (currently E). Also, the legend is incorrect here too, as panels E, F and G are listed as A, B and C.

Response: Corrected appropriately

Question 10. It is claimed that complementation rescued the invL complemented strain. However, comparison of the complemented mutant against the mutant showed no statistically significant change.

Response: The description of the data has been clarified appropriately.

Question 11. Is there any evidence that InvL antibodies are observed in long-term human infections?

Response: No published data. I accept that this is beyond the scope of this paper.

Question 12. In Figure 5 each data point is labelled as representing an individual mouse, but only single dots are visible. Furthermore, the horizontal line is not really a mean as it connects the means from each time point but actually is mostly interpolated data.

Response: This has been modified appropriately.

Question 13. Figure 6. For panels A, B and C, please label the starting x axis value (1 not 0). Also, there is meant to be a limit of detection dotted line, but I can't see it. Is it 1 CFU so lost in the base line?

Response: This has been modified appropriately.

Question 14. In the description of Figure 6 on lines 276, it states that *K. pneumoniae* significantly decreased *A. baumannii* CFU in the lungs relative to mock infected and untreated groups; however, this only appears to be significant at 1 day p.i. and not 2 days p.i.

Response: Clarified appropriately.

Question 15. It is speculated that one reason for the reduced survival of *A. baumannii* and *K. pneumoniae* in polymicrobial infections may be the presence of a T6SS. This should be tested directly with an *A. baumannii* tssM mutant.

Response: This was not attempted but accept that this is beyond the scope of the manuscript.

Question 16. I think it would be helpful to have some comments/discussion on, why a tlr4 knockout model might be a more relevant model than wt mice when both humans and mice have tlr4. Do you expect the model would still have relevance for analysis of bacterial mutants with altered LOS? 7-day model using Klotho mice (longer term model, higher dose, older strain). Low dose infections together with gastric mucin (10⁵ CFU, acute model, with older strain, although).

Response: The increased discussion on these points is a good addition.

Additionally, on further reading of the manuscript, I think that the speculation on the colistin and tigecycline treatments selecting for persisters should be tempered. As the authors define persisters as those cells "that become tolerant to antibiotics despite undergoing no genetic changes" this has not been shown in this paper. While a significant portion of the bacterial population is killed following colistin and tigecycline treatment, and the numbers of surviving bacteria stay static for the next few days, this is not necessarily because they are persisters. Firstly, the genetics of the surviving cells have not been investigated (whole genome sequencing could be performed), so there may have been selection for particular mutants. Secondly, while lack of growth in the face of antibiotic treatment in vitro (in the absence of immune system) is highly indicative of persisters (as the cells are neither dying nor replicating) the interpretation in an animal is more complex as they may be growing (due to resistance) but at a rate equal to their removal by immune system killing.

Finally, a very minor point that all graph titles are underlined but Fig. S2 panel C is not (the 7dpi title).

Reviewer #2

(Remarks to the Author)

In this revised submission, the authors have made substantial efforts to address my initial concerns by conducting additional experiments and manuscript modifications. These modifications have adequately addressed my concerns. Thanks for your efforts.

Reviewer #3

(Remarks to the Author)

The authors have adequately addressed all my comments.

Dear Editors,

Please find attached the revised version of our manuscript for consideration for publication by Nature Communications. We have carefully considered the reviewers concerns and modified our manuscript accordingly.

Note to Reviewers:

In line with recommendations in the “Reporting Summary,” we have now included a source data file. Where relevant, we have included n, test statistics, *P* values, and the flow cytometry gating strategy according to journal guidelines, and this has been referenced in the legends and in the Methods section.

We have now included a Data Availability Statement to address journal requirements.

We have now included an Ethics Statement to address journal requirements.

While putting together the source data for the manuscript, we noted the following:

1. In Figures 2B and 2E, for flow cytometry analyses of 2 dpi, one of the biological replicates was duplicated in the data set in the first submission. We have removed the duplicate and updated the Figure (See Fig. 2) and text (See lines 160-182) accordingly. This led to a loss of statistical significance between AMs of *tlr4* mutant mice infected with 10^5 and 10^8 bacteria and between PMNs of WT and *tlr4* mutant mice infected with 10^8 bacteria at this timepoint. Importantly, this update does not change the conclusion of the section or manuscript.
2. Missing values in the data for Table S2 were identified and corrected, changing 7 of the values in the table (4 hpi – WT – Mock – IL-23; 4 hpi – *tlr4* mutant – 10^5 – IL23; 4 hpi – *tlr4* mutant – mock – IL23; 4 hpi – WT – 10^5 – TNF-a; 2 dpi – WT – 10^5 – TNF-a; 2 dpi – WT – 10^5 – TNF-a – *lfn-B*; 7 dpi – WT – 10^9 – *lfn-g*). This has been corrected (See Table S2). Notably changes did not affect conclusions of the manuscript.
3. There was one typo in the 10^8 *tlr4* mutant infection data in Figure 1. This has been corrected, and conclusions have not been affected.
4. The wrong *gfp* construct was included in the manuscript. The correct one has been inserted and conclusions have not been changed.

Response to Reviewers:

Reviewer #1 (Remarks to the Author):

This paper describes the development of a long-term *Acinetobacter baumannii* mouse model (~3 week infections) using two recent *A. baumannii* clinical isolates. Currently the majority of *A. baumannii* mouse disease models last only a few days, require high doses and are often using lab-adapted strains. The model described here is interesting as it utilises lower doses than many other *A. baumannii* mouse models and gives long-term bacterial colonisation. Utilisation of the new model allowed identification of a virulence factor that appears to be specific for long-term carriage. Furthermore, the model was used to assess polymicrobial interactions and looks useful for testing efficacy of antibiotics over longer time periods. The work will be of interest to those that work with *A. baumannii* and may be relevant to development of mouse models for some other pathogens, although that is not shown in this work.

Major Points

1. How phylogenetically similar are the two clinical strains that were tested (G636 and G654). Are they diverse enough to give insight into general applicability?

We have now reviewed the genetic similarity of G636 and G654 by comparing MLST sequence typing, and, while these strains are distinct, they have the same sequence type (208) and were both isolated from Georgia hospitals within a 2-year period. We have therefore included a modern Iranian isolate, strain G803 in this study that has a distinct sequence type, 1624. We have updated the manuscript to include MLST numbers of strains tested in the long-term infection model (See Table S1).

2. A number of times it is noted that high dose models are likely not clinically relevant, although the natural clinically relevant dose is difficult to predict and may fluctuate widely, especially where infections may follow direct inoculation of indwelling devices that may be contaminated with a biofilm. It initially appears surprising that in *tlr4* mutant mice the high dose is cleared more rapidly than the low dose (e.g. high doses mostly cleared by 7-10 days and low doses not often cleared even after 19 days). It would be important to show a wider range of doses to compare when this long-term survival becomes most marked and when, or if, the *tlr4* mice can completely clear the infection. Specifically, would a lower dose of 10^3 give increased, reduced or similar levels of organisms over the 19 days? Is 10^5 the optimal dose in this model? Understanding how sensitive the model is to dose will guide more widespread and appropriate uptake.

We screened a range of doses of G636, from 10^3 - 10^8 , in *tlr4* mutant mice, but did not include this in the initial manuscript. We agree with the reviewer's suggestion though that showing this data is important, so we have included the results of this screen and have further discussed this in the Results section (See Figure S3 and lines 126-132). We have also included one experiment where we investigated later timepoints with the 10^5 inoculum in C3H/HeJ mice with strains G636 and G654 to assess when mice clear infection (See Figure S1). Following 21 dpi, infection became inconsistent however, which is why we did not exceed this timepoint in any following experiments.

3. For the detection of intracellular G636, did the GFP-labelled strain grow at the same rate as the WT G636 strain?

To address this concern, we have included *in vitro* growth curves of G636 wild-type and *gfp*-expressing strains, demonstrating no differences in growth (See Figure S2).

4. Are there only 2 mice for the G654 initial WT infections (Fig. 1)?

We apologize for any confusion regarding this figure. Mouse numbers are now further clarified in the Source Data file. For these initial screenings, groups of 2-3 mice were sacrificed at each timepoint.

5. Fig 2. In panel A. Is 10^5 vs 10^8 for TLR4 mutant not significant? Please check.

To address this comment, we have gone back and reanalyzed all original flow data and statistical comparisons. This comparison is indeed significant. We have updated the manuscript with the new figure and have updated the text accordingly (See lines 160-182). We thank the reviewer for catching this.

6. The only significant difference in cytokines between low dose infections in WT and low dose infections in *tlr4* mice was GM-CSF. It would be very interesting to test the outcome of blocking the action of this cytokine in WT mice.

We appreciate this comment from the reviewer, and we agree that it would be interesting to assess the role of GM-CSF during *A. baumannii* infection. We feel that identifying the role of specific cytokines in the chronic infection model falls outside of the scope of the current manuscript. Ongoing work in the lab will be aimed at addressing specific immunological mechanisms underlying clearance vs. long-term infection in this model.

7. Fig. S2 has panel B in the image labelled 2 h pi when it is 2 day pi

We have corrected it in the updated figure.

8. In figure S4 are differences for *invL* mutant significant at 14 days?

Because we were showing just one example at 1 and 14 dpi of why we chose to perform more replicates with the *invL* mutant, we did not include statistics here. Statistics are shown though in the more detailed analysis with 3 pooled biological replicates presented in Fig. 4.

9. Fig. 4, panel G not labelled correctly (currently E). Also, the legend is incorrect here too, as panels E, F and G are listed as A, B and C.

We appreciate the reviewer finding this mistake. It has been corrected in the updated figure.

10. It is claimed that complementation rescued the *invL* complemented strain. However, comparison of the complemented mutant against the mutant showed no statistically significant change.

We appreciate the reviewer's concern, as our conclusion required more clarity. Full complementation was not noted at every timepoint as pointed out by the reviewer. We have now updated the text with more specificity regarding statistical significance (See lines 245-253).

11. Is there any evidence that *InvL* antibodies are observed in long-term human infections?

We have performed a literature search for *InvL*, and did not find any publications indicating that *InvL* antibodies are found in any human infections. This is likely because our team only recently discovered *InvL* in 2022 (Jackson-Litteken and Di Venanzio et al., *mBio*, 2022). Future work will be aimed at addressing this question.

12. In Figure 5 each data point is labelled as representing an individual mouse, but only single dots are visible. Furthermore, the horizontal line is not really a mean as it connects the means from each time point but actually is mostly interpolated data.

We appreciate the reviewer finding this mistake. It has been corrected in the text here (See Fig. 5), as well as for Fig. S10.

13. Figure 6. For panels A, B and C, please label the starting x axis value (1 not 0). Also, there is meant to be a limit of detection dotted line, but I can't see it. Is it 1 CFU so lost in the base line?

We have updated the figure to include 1 as the starting point on the X-axis (See Fig. 6). We have also added a LOD line to this figure (10 CFU). We have also included the LOD line to other relevant figures.

14. In the description of Figure 6 on lines 276, it states that *K. pneumoniae* significantly decreased *A. baumannii* CFU in the lungs relative to mock infected and untreated groups; however, this only appears to be significant at 1 day p.i. and not 2 days p.i.

We agree that this needed clarification. We have addressed this in the text, specifically indicating that this significant difference is at 1 dpi and not 2 dpi. We have also added text to increase specificity about changes to *K. pneumoniae* CFU as well (See lines 321-329).

15. It is speculated that one reason for the reduced survival of *A. baumannii* and *K. pneumoniae* in polymicrobial infections may be the presence of a T6SS. This should be tested directly with an *A. baumannii* tssM mutant.

We agree with the reviewer that the possibility that *A. baumannii* and *K. pneumoniae* T6SS-mediated competition is occurring *in vivo* is a fascinating facet that can be investigated in the context of this model. This possibility was listed as a discussion point/speculation along with others including competition for nutrients or a differential immune response elicited by the combination of *A. baumannii* and *K. pneumoniae* that could be leading to the observed phenotype. To better understand our phenotype from these co-infections, testing of all these possibilities is warranted. As such, we will be pursuing this in future work, but we believe that assessing the role of each of these is beyond the scope of the current manuscript.

16. I think it would be helpful to have some comments/discussion on,
a. Why a *tlr4* knockout model might be a more relevant model than wt mice when both humans and mice have *tlr4*. Do you expect the model would still have relevance for analysis of bacterial mutants with altered LOS?

The primary reason that we pursued an immunocompromised mouse model is that WT mice are resistant to *A. baumannii* infection, leading to the requirement of using large inoculums that are quickly cleared. This method provides a very short window to investigate bacterial pathogenesis. We agree with the reviewer here though that an extensive discussion on why we used *tlr4* mutant mice and the limitations of this model is warranted in in this manuscript. To

begin to address this, we have clarified the potential clinical significance of a *tlr4* mutant model in the Discussion, highlighting the several studies that have demonstrated links between alterations in human TLR4 and *A. baumannii* pathogenicity (See lines 345-356). Importantly though, as the reviewer pointed out, while primarily immunocompromised people are infected by *A. baumannii*, changes in TLR4 are likely not the mechanism underlying all infections. We have therefore included a thorough discussion of this limitation and others, including its limitation in studying mutants with altered LOS, that may not be recognized by TLR4. (See lines 357-389).

b. 7-day model using Klotho mice (longer term model, higher dose, older strain).

To our knowledge, Klotho mutant mice have been employed with *A. baumannii* in an intravenous sepsis model (Sato et al., *Front Immunol*, 2020). We agree though that it would be interesting to assess if these mice are more susceptible to long-term pulmonary infection following intranasal or intratracheal inoculation. Use of these mice would have some limitations as, similar to mutations in *tlr4*, mutations in *klotho* and associated effects are not the likely cause of most *A. baumannii* infections. However, if Klotho mutant mice have extended pulmonary infection in response to lower inoculums similar to *tlr4* mutant mice, we do believe that a Klotho mutant model and the chronic model presented here could be used in conjunction to compare/confirm phenotypes between long-term models. We have therefore included this as a possibility in the Discussion (See line 386-389).

c. Low dose infections together with gastric mucin (10^5 CFU, acute model, with older strain, non-respiratory).

Gastric mucin has been used in several papers in intraperitoneal models to increase murine susceptibility to *A. baumannii* infection. It has also been used in one study, to our knowledge, for a pulmonary infection model, but infection was only followed up to 3 dpi (Tang et al., 2012). We agree with the reviewer here that it is possible to use this method in the pulmonary infection model to assess if this method can extend infection, similar to what has been done with other pathogens. We have included this as a possibility in the discussion (See lines 386-389).

Minor Points

1. Fig. 1 legend “Each data point indicates and individual” should read “Each data point indicates an individual”

This has been corrected (See line 688).

2. Fig. S1 legend “Following, bacterial CFU” should read “Bacterial CFU”

This has been corrected (See line 783).

3. Figure 5 legend, “were treated intraperitoneally treated with PBS” should read “were treated intraperitoneally with PBS”

This error has been corrected in this figure legend (See line 742), as well as the legend for Figure S10 (See line 870).

4. Figure 6 legend, “were infected on *K. pneumoniae*” should read “were infected with *K. pneumoniae*”

This error it has been corrected (See line 757).

5. Line 280 should refer to Fig. S7D-F not just D-E

We thank the reviewer for catching this error, and it has been corrected (See line 326).

Reviewer #2 (Remarks to the Author):

In their manuscript, Dr. Jackson-Litteken and colleagues reported their efforts to develop a mouse model of chronic respiratory *Acinetobacter baumannii* (Ab) infection using *tlr4* mutant mice (C3H/HeJ) and a low bacterial inoculum. Using this model, the authors investigated the potential role of adhesin *InvL* as the key virulence factor in establishing chronic infection. Attempts have also been made to demonstrate the potential utility of this model for evaluating antibiotic efficacy and studying polymicrobial infections.

The mouse model reported in this manuscript is new and interesting. If validated, it will complement other available mouse models for infection pathogenesis studies and potential therapeutic evaluation. However, several experiments (such as the mutant study and the polymicrobial infection) were largely descriptive and more in-depth studies will be needed. In particular, the robustness of the model and the role of *InvL*-mediated persistent infection have suffered from the high variations in the tissue bacterial burdens between individual mice, even though statistical significance is achieved in some instances.

1. The authors tried to demonstrate that the ability to establish the chronic infection is somewhat associated with the modern isolates by using ATCC19606 as a control. To support this assumption, additional old strains will need to be included since ATCC19606 is a low-virulence strain in mice and may not be an ideal representation of the old strains.

We appreciate this concern brought up by the reviewer. To address this point, we performed an additional experiment where we screened *Ab17978* at the 10^5 inoculum in *tlr4* mutant mice in a 7-day infection experiment (See Fig. S4). The reviewer’s assertion that other older strains should have been tested was indeed correct and necessary, as we found that *Ab17978*, unlike *Ab19606*, was able to establish infection out to the latest timepoint tested, and infection kinetics to this point reflected those of the modern respiratory isolates tested. We have updated the text and clarified our conclusions accordingly (See lines 142-151).

2. Mechanisms of chronic infection in this model: From the pulmonary cellular and cytokine response data, the authors have ruled out their contribution to the development of chronic infection, which leaves its mechanisms to be further determined. It will be interesting to determine the potential role of PMNs in the development of the chronic infection in this model.

We agree that it would be interesting to identify the role of PMNs in the chronic infection model. In order to address this comment and begin to understand the role of PMNs, we performed an

experiment where, prior to infection, we rendered the mice neutropenic by cyclophosphamide treatment via a well-established protocol by the Luis Martinez lab (Manepalli et al., 2013). In this model, mice are treated with 300 mg/kg cyclophosphamide. At 3 days post-treatment, as few neutrophils are detected between 3-4 days following cyclophosphamide treatment, mice were infected with 10^5 G636. This infection resulted in initially increased CFU by 100-fold relative to the chronic infection model, indicative of a potentially important role for PMNs in controlling infection (See Fig. S6 and lines 183-192). As expected, as neutropenia wanes in this model following 3-4 days post-cyclophosphamide treatment, bacterial numbers approached similar amounts to that seen in the chronic infection model at later timepoints. While further work is necessary to mechanistically determine the role of neutrophils in preventing bacterial expansion in the chronic infection model, this experiment provides the initial indication that PMNs are still important in infection control even in the absence of a functional copy of *tlr4*.

3. Histopathology analysis (Fig. 3): Was the histologic analysis and scoring done by a laboratory animal pathologist? Some chronic histopathologic changes (such as squamous epithelium metaplasia and BALT formation etc.) were observed as early as 4 hpi. Any reasons for this?

The original lung sections were not reviewed by a licensed animal pathologist, but rather by authors of the manuscript. To address this concern, we have included two additional biological replicates of this experiments, and all samples were then reviewed and scored by the director of the Washington University Department of Comparative Medicine Diagnostic Animal Laboratory, Dr. Leslie D. Wilson, PhD, DVM, DACVP, who is a board-certified veterinary pathologist licensed through the American College of Veterinary Pathologists (ACVP) and who is now included as an author on this manuscript.

Dr. Wilson found that there are indeed chronic histopathological changes early during infection in infected, as well as mock animals, as was found in the initial analysis. An extensive review of the literature on C3H/HeJ mice has revealed multiple papers indicating that mice of this strain background i) have abnormal lung development, ii) are commonly colonized by environmental Gram-negative bacteria found in the environment of mouse breeding/housing as the result of the *tlr4* mutation, and iii) spontaneously develop emphysema. These are potential reasons for the altered pathologies that were noted by the reviewer. We have included updated results for the new analysis and included a thorough discussion of the C3H/HeJ strain in the manuscript (See lines 211-231).

Importantly, however, when combining all results reviewed by Dr. Wilson, infection still resulted in significant increases in alveolitis and attenuated airway epithelium relative to mock infection (See Fig. 3). These results, particularly the attenuated airway epithelium later during infection, are consistent with persistent, chronic infection in our model.

4. The role of InvL in the persistence of Ab in the lung: The data on the mutant and complementation strain were highly variable between individual mouse. Additional in vitro cell culture and in vivo data to demonstrate the increased adhesin expression and bacterial adhesion to the respiratory epithelial cells in the chronic infection will be needed to support the overall hypothesis.

We appreciate this concern from the reviewer and agree that there is variability. Unfortunately, variability is very common with murine infection models of *Acinetobacter baumannii* and other

respiratory pathogens (e.g., *Pseudomonas aeruginosa* and *Staphylococcus aureus*) with bacterial numbers in acute models at 24-36 hpi often having a range of $\sim 2 \log_{10}$. With extended time in a biological system and lower doses, we expect increased variability between individual mice. Studies with other bacteria in chronic pneumonia models support this expectation. For example, multiple studies show that lower doses of *Mycobacterium tuberculosis* result in significantly higher variability in recoverable bacterial CFU at later timepoints during infection, and it has been suggested this is more relevant to the variability of human infection, as mouse studies are often confounded by unrealistically high inoculums (Saini et al., *Tuberculosis*, 2012; Karp et al., *Immunol Rev*, 2015; Plumlee et al., *Cell host & microbe*, 2021).

Regarding the role of InvL, while we haven't specifically explored the interaction of the G636 *invL* mutant with epithelial cells in this manuscript, we have shown that recombinant InvL from *A. baumannii* strain UPAB1, which shares 98.8% homology with G636 InvL at the full protein level and 100% homology in the lectin-binding domain, binds to multiple extracellular matrix proteins that are known to be expressed in the lung in a previous publication, highlighting its adhesive properties (Jackson-Litteken and Di Venanzio et al., *mBio*, 2022). We have additionally shown significantly reduced adherence of the UPAB1 *invL* mutant to multiple epithelial cell lines in the previous manuscript. These observations both align with the phenotype observed here. It is important to note that we included this figure as a proof of concept that the chronic infection model can be used to attempt to identify bacterial virulence factors important for later stages of infection. This was one of three examples of how this model could potentially be used in the future, and mechanistically assessing the various phenotypes observed in each of these proofs of concept would not be feasible to present in one manuscript, although it is clearly warranted as pointed out by the reviewer. As such, we are currently pursuing the mechanistic role of InvL in pulmonary pathogenesis in an upcoming manuscript, and it is the subject of an already submitted grant application.

5. Please indicate the detection limits for cytokines and all the bacterial burden assays.

The limits of detection have now been included in (Table S2).

6. Fig. 1: The legend indicates that the samples were collected every 3 days (Ln 575), which appears to be inconsistent with the plots in the figure. Also, please correct a typo on Ln 576 (and to an).

We apologize for this confusion. This discrepancy came from using the "staggered" setting for data points in the Prism software. This was initially used because the only alternative setting, "aligned," sometimes led to overlap of datapoints. We agree with the reviewer though that the staggered setting makes it difficult for the reader to determine when timepoints are taken. We have therefore updated Figure 1 to show the "aligned" setting for clarity (See Fig. 1). We have also corrected the noted typo (See line 688). We thank the reviewer for catching this.

7. As anticipated, the mice inoculated with a high dose of Ab have cleared the infection rapidly (Fig. 1). However, the number of PMNs in the lung of WT mice (Fig. 2F) remains significantly elevated at dpi 7. This observation is inconsistent with published data although the different Ab and mouse strains were used in these studies. Some discussions on this will be useful.

We had not considered neutrophil kinetics in the lungs following high-dose infection in WT (C3H/HeN) mice, and we thank the reviewer catching the differences and suggesting some discussion. We have done an exhaustive literature search on murine infections with high dose bacteria and reported neutrophil kinetics. This has been included now in the manuscript and compared to our results (See line 169-182).

8. Fig. 4: A large number of mice were used in this experiment. The legend indicates that the results shown are from 3 independent experiments. Can the authors clarify if the data shown in the figure are one of the 3 independent experiments or the results of 3 pooled experiments? If the latter, it will be important to show if the result from each individual experiment supports the conclusion, given the high intragroup variation in some groups.

We have reanalyzed the figure data and we are presenting only pooled, matched data from 3 independent experiments with 4-5 mice/timepoint/experiment together in Fig. 4. We acknowledge that there is high variability and up to 15 mice/timepoint, so, as requested by the reviewer, we are now also showing these 3 independent experiments individually in Fig. S9. Results were consistent across replicates, as minimal changes between WT/complemented mutant and mutant strains were noted early during infection. By 21 dpi, all mice in 2/3 mutant infected replicates had completely cleared infection, whereas WT and complemented mutant infected mice in each replicate still maintain detectable CFU.

9. Some discussions on how this model reflects the human infection are needed since the long clinical duration of the Ab infection could largely be due to the antibiotic resistance of the Ab strain and may not be associated with a specific immune factor such as TLR 4.

We agree with the reviewer that infection is not always due to one specific immune factor such as TLR4. We have therefore included a section in the Discussion highlighting the potential relevance of a *tlr4* mutant infection model and its limitations (See line 345-389). In this section, we acknowledged what the reviewer's point regarding infection being associated with multidrug resistance. We also stress the importance of identifying other models that can be used in conjunction to study long-term virulence. Importantly, while this model has limitations, it still provides an important tool, as it the first enabling investigation of *A. baumannii* mechanisms of virulence and drug resistance over the course of weeks of infection, rather than days.

Reviewer #3 (Remarks to the Author):

In this manuscript, the authors report the development and application of a novel chronic murine model of pulmonary *Acinetobacter baumannii* infection. Current available murine pulmonary *Acinetobacter* models are all based on acute infection that require a very high bacterial inoculum (10^8 - 10^9). This artificial host-pathogen interaction might skew many aspects of *Acinetobacter* pathogenesis. The authors reported a very exciting chronic *Acinetobacter* murine model allows for studying late virulence factors, long-term antibiotic treatments, and polymicrobial infections. Overall, the study is well designed and presented. Here are some minor comments:

1. Based on the Fig. 4E-F bacterial burden data, the authors concluded that InvL is dispensable

in the acute infection. Does the *invL* mutant also kill the wt and *tlr4* mutant mice similar to G636 in the acute model? Please correct the legend describing acute infection Fig. 4E-F.

For the acute infection model, we did not employ a lethal dose of G636. Therefore, it was not possible to determine mortality with WT, *invL* mutant, and complemented strains in these experiments. In our experience, in cases where no significant difference in bacterial CFU is identified at 24 hpi, increasing the dose does not tend to reveal a phenotype. This is likely due to the extremely high inoculums masking potential defects. We have also corrected the legend (See Fig. 4). We thank the reviewer for catching this mistake.

2. Please expand the discussion regarding the relevance of this novel chronic infection model to clinical disease (e.g. immune response, bacterial persistence, etc.). Also, provide insight into the potential limitation and consideration when using this long-term model, since TLR4 plays an important role in bacterial recognition and innate immunity against *A. baumannii*.

We agree with the reviewer that these points need to be addressed when implementing a new murine model. We have therefore expanded on the discussion the relevance of TLR4 in *A. baumannii* human infection, as well as discussed the limitations of the model (See line 345-389).

3. Please confirm the label of Figure S2 B is 2 dpi.

We have corrected this and thank the reviewer for catching this oversight.

REVIEWERS' COMMENTS

Reviewer #1 (Remarks to the Author):

This is a review of revision one of the article by Jackson-Litteken et al.

During initial review I asked for numerous revisions. Mostly, these have been addressed adequately, and I believe the manuscript is much improved. I have noted each initial question and my interpretation of the authors responses below.

I have one new point that I believe should be addressed, and this is noted after the list of questions and responses.

As noted in my original review, I believe this work will be of interest to those that work with *A. baumannii* and may be relevant to development of mouse models for some other pathogens, although that is not shown in this work. Utilisation of the new model allowed identification of a virulence factor that appears to be specific for long-term carriage. Furthermore, the model was used to assess polymicrobial interactions and looks useful for testing efficacy of antibiotics over longer time periods.

Question 1. How phylogenetically similar are the two strains that were tested (G636 and G654)?

Response: The two initial strains used were indeed highly phylogenetically similar. A new strain with a different sequence has now been included. While it would have been more comprehensive to determine whole genome sequences (rather than just MLST sequence types) and place all the strains (including the ATCC strains) on a whole genome phylogeny, the added strain addresses my main concern here.

Thank you.

Question 2. Why was 10^5 CFU chosen? Would a lower dose of 10^3 give increased, reduced or similar levels of organisms over the 19 days? Is there a linear response or is 10^5 the optimal dose in this model?

Response: The authors have now added in data for different starting doses, which appropriately addresses this concern.

Thank you.

Question 3. For the detection of intracellular G636, did the GFP-labelled strain grow at the same rate to the WT G636 strain?

Response: A full growth curve has now been added showing the two strains do grow at similar rates

Thank you.

Question 4. Are there only 2 mice for the G654 initial WT infections (Fig. 1)?

Response: Now clarified

Thank you.

Question 5. Fig 2. In panel A. Is 105 vs 108 for TLR4 mutant not significant? Please check.

Response: Was significant and changed appropriately

Thank you.

Question 6. The only significant difference in cytokines between low dose infections in WT and low dose infections in tlr4 mice was GM-CSF. It would be very interesting to test the outcome of blocking the action of this cytokine in WT mice.

Response: While this was not performed, I accept the suggestion that this may be beyond the scope of this manuscript

Thank you.

Question 7. Fig. S2 has panel B in the image labelled 2 h pi when it is 2 day pi

Response: Corrected appropriately

Thank you.

Question 8. In figure S4 are differences for invL mutant significant at 14 days?

Response: Clarified appropriately

Thank you.

Question 9. Fig. 4, panel G not labelled correctly (currently E). Also, the legend is incorrect here too, as panels E, F and G are listed as A, B and C.

Response: Corrected appropriately

Thank you.

Question 10. It is claimed that complementation rescued the invL complemented strain. However, comparison of the complemented mutant against the mutant showed no statistically significant change.

Response: The description of the data has been clarified appropriately.

Thank you.

Question 11. Is there any evidence that InvL antibodies are observed in long-term human infections?

Response: No published data. I accept that this is beyond the scope of this paper.

Thank you.

Question 12. In Figure 5 each data point is labelled as representing an individual mouse, but only single dots are visible. Furthermore, the horizontal line is not really a mean as it connects the means from each time point but actually is mostly interpolated data.

Response: This has been modified appropriately.

Thank you.

Question 13. Figure 6. For panels A, B and C, please label the starting x axis value (1 not 0). Also, there is meant to be a limit of detection dotted line, but I can't see it. Is it 1 CFU so lost in the base line?

Response: This has been modified appropriately.

Thank you.

Question 14. In the description of Figure 6 on lines 276, it states that *K. pneumoniae* significantly decreased *A. baumannii* CFU in the lungs relative to mock infected and untreated groups; however, this only appears to be significant at 1 day p.i. and not 2 days p.i.

Response: Clarified appropriately.

Thank you.

Question 15. It is speculated that one reason for the reduced survival of *A. baumannii* and *K. pneumoniae* in polymicrobial infections may be the presence of a T6SS. This should be tested

directly with an *A. baumannii* tssM mutant.

Response: This was not attempted but accept that this is beyond the scope of the manuscript.

Thank you.

Question 16. I think it would be helpful to have some comments/discussion on, why a tlr4 knockout model might be a more relevant model than wt mice when both humans and mice have tlr4. Do you expect the model would still have relevance for analysis of bacterial mutants with altered LOS? 7-day model using Klotho mice (longer term model, higher dose, older strain). Low dose infections together with gastric mucin (10⁵ CFU, acute model, with older strain, although).

Response: The increased discussion on these points is a good addition.

Thank you.

Additionally, on further reading of the manuscript, I think that the speculation on the colistin and tigecycline treatments selecting for persisters should be tempered. As the authors define persisters as those cells “that become tolerant to antibiotics despite undergoing no genetic changes” this has not been shown in this paper. While a significant portion of the bacterial population is killed following colistin and tigecycline treatment, and the numbers of surviving bacteria stay static for the next few days, this is not necessarily because they are persisters. Firstly, the genetics of the surviving cells have not been investigated (whole genome sequencing could be performed), so there may have been selection for particular mutants. Secondly, while lack of growth in the face of antibiotic treatment in vitro (in the absence of immune system) is highly indicative of persisters (as the cells are neither dying nor replicating) the interpretation in an animal is more complex as they may be growing (due to resistance) but at a rate equal to their removal by immune system killing.

We agree with the reviewer and have now referred to these cells as “possible” persisters. Thank you.

Finally, a very minor point that all graph titles are underlined but Fig. S2 panel C is not (the 7dpi title).

We have now fixed this. Thank you.

Reviewer #2 (Remarks to the Author):

In this revised submission, the authors have made substantial efforts to address my initial concerns by conducting additional experiments and manuscript modifications. These modifications have adequately addressed my concerns. Thanks for your efforts.

Thank you.

Reviewer #3 (Remarks to the Author):

The authors have adequately addressed all my comments.

Thank you.